# Revolutionary Cancer Therapy for Personalization and Improved Efficacy: Strategies to Overcome Resistance to Immune Checkpoint Inhibitor Therapy

**DOI:** 10.3390/cancers17050880

**Published:** 2025-03-04

**Authors:** Saud Almawash

**Affiliations:** Department of Pharmaceutics, College of Pharmacy, Shaqra University, Shaqra 11961, Saudi Arabia; salmawash@su.edu.sa; Tel.: +966-565552648

**Keywords:** immunotherapy, immune-checkpoint inhibitors, PD-1/PD-L1, CTLA-4, monoclonal antibodies, combination therapies, immunotherapy resistance, biomarkers, tumor mutational burden, artificial intelligence

## Abstract

Cancer remains a major global health issue, causing millions of deaths each year. Immunotherapy, a treatment approach that uses the body’s immune system to fight cancer, has transformed cancer care. One of the most successful immunotherapy strategies involves immune checkpoint inhibitors, which help immune cells attack cancer more effectively. However, not all patients respond well to these treatments; some develop resistance or experience severe side effects. Scientists are working to overcome these challenges by combining different therapies, developing new immune-targeting drugs, and identifying biological markers that can predict which patients will benefit the most. This review explores how immune checkpoint inhibitors work, why resistance occurs, and how researchers are developing new ways to improve treatment effectiveness. By understanding these factors, we can enhance cancer treatment and improve outcomes for more patients, ultimately advancing the field of oncology.

## 1. Introduction

Cancer remains a significant public health issue worldwide, standing as a primary contributor to global mortality, accounting for approximately 10 million fatalities in 2020. It is the primary contributor to global mortality and a leading cause of morbidity and mortality in numerous countries [1]. However, there have been notable advancements in cancer research and treatment in recent years, particularly in personalized medicine and therapeutics [2]. One of the most transformative advancements in cancer therapy is immunotherapy, which uses the immune system to combat cancer cells. Currently, immunotherapy is used as a monotherapy or in conjunction with numerous conventional therapies, including chemotherapy (CT) and radiotherapy (RT), and has demonstrated remarkable efficacy in treating various cancers [3].

Decades of T-cell biology and tumor immunology research have shown that cancer immunotherapy is now a standard treatment for numerous types of tumors [4]. This constituted a substantial advancement in oncology [5], with many cancer types demonstrating prolonged therapeutic responses to immunotherapy [6,7,8,9]. The cancer immune cycle is initiated by releasing antigens from cancerous cells. The presentation of these antigens on antigen-presenting cells (APCs) to antigen-specific T cells by major histocompatibility complexes (MHCs) leads to the priming and activation of T cells [10]. However, as these T cells have the potential to damage healthy cells during activation and proliferation, they are regulated by immune-checkpoint (IC) pathways that modulate their responses to prevent autoimmunity and maintain immune homeostasis [11].

Programmed death-1 (PD-1), which is expressed on T cells, B cells, and myeloid cells, and its ligands programmed death-ligand 1 (PD-L1) and programmed death-ligand 2 (PD-L2), which are expressed on tumor cells and APCs, inhibit T-cell activation and proliferation when bound, reduce cytokine production, and impair cytotoxic functions (Figure 1) [12,13]. Cytotoxic T-lymphocyte-associated antigen 4 (CTLA-4), which is primarily expressed in T cells, competes with Cluster of Differentiation 28 (CD28) to bind CD80/CD86 on APCs, providing an inhibitory signal that reduces T-cell activation and proliferation (Figure 1). This is a crucial process in the early stages of T-cell activation in lymphoid organs [14]. However, tumors exploit these pathways to evade immune surveillance, facilitating immune escape and tumor progression. Lymphocyte-activation gene 3 (LAG-3), which is expressed on activated T cells, regulatory T cells (Tregs), natural killer (NK) cells, B cells, and dendritic cells, binds to major histocompatibility complex (MHC) Class II molecules and negatively regulates T-cell proliferation, activation, and homeostasis. In addition, it plays a role in Treg function and the maintenance of immune tolerance [15]. T cell immunoglobulin and ITIM domain (TIGIT), which is expressed on T cells, NK cells, and some regulatory T cells (Tregs), binds to CD155 and CD112, inhibiting T-cell and NK cell cytotoxicity, promoting T-cell exhaustion, and supporting the immunosuppressive function of Tregs [16]. B7 homolog 3 protein (B7-H3), which is expressed in T cells, NK cells, and various non-immune cells, inhibits T-cell effector functions, contributes to immune evasion, and is often overexpressed in tumors (Figure 1) [17]. V-domain immunoglobulin suppressor of T-cell activation (VISTA), which is expressed on myeloid cells and T cells, binds to PSGL-1 and negatively regulates T-cell activation and cytokine production. Furthermore, it plays a role in maintaining peripheral tolerance and suppressing antitumor immunity [18]. B and T lymphocyte attenuator (BTLA), expressed on T cells, B cells, and some myeloid cells, binds to herpes virus entry mediator (HVEM). This binding delivers inhibitory signals to T cells, reducing their proliferation and cytokine production. BTLA plays a role in maintaining immune homeostasis and tolerance. A comprehensive understanding of these pathways is essential for developing effective immunotherapies, such as immune checkpoint inhibitors (ICIs), which aim to block these checkpoints and restore antitumor immunity [19].

Tumor cells have evolved mechanisms to evade immune detection via immune-checkpoint (IC) molecules. However, understanding the infiltration of immune cells into the tumor microenvironment (TME) is crucial for enhancing response rates and developing novel cancer immunotherapy techniques. Over 10 ICs have been identified in the field of immunology, with the PD-1/PD-L1 axis being extensively studied [20,21]. In the context of TME, the PD-1/PD-L1 pathway prominently emerges as the principal IC [22]. This pathway ordinarily plays a crucial role in maintaining immune homeostasis and in preventing excessive immune activation. However, malignant cells have evolved to hijack this pathway, distorting its physiological functions and establishing an immunosuppressive environment that facilitates immune evasion. Monoclonal antibodies (mAbs) have emerged as highly effective immunotherapeutic agents to combat cancer. mAbs can recognize and bind to singular epitopes with high precision. This exceptional specificity renders mAbs invaluable for therapeutic purposes, as they can be customized to selectively target particular molecules, disrupting pathways that contribute to disease progression [23]. This fundamental principle underscores the applicability of ICIs in cancer immunotherapy. By precisely targeting and impeding immune checkpoints such as CTLA-4 and PD-1, immune-checkpoint inhibitors (ICIs) rekindle immune activity, empowering them to discern and eliminate cancerous cells [24]. This novel approach represents a promising frontier in the ongoing fight against cancer.

Immunotherapy, despite eliciting sustained immune responses that can persist beyond therapy cessation, can lead to prolonged antitumor effects and overall survival, but some patients do not experience a significant reduction in tumor burden [25]. ICI therapy benefits only 20–40% of patients, and some may experience severe immune-related adverse events (irAEs). The inhibition of ICs triggers irAEs, a range of autoimmune reactions at specific immune system sites, making it crucial to develop strategies to overcome treatment resistance [2,26].

This review will focus on the most widely recognized ICs and the effect of their inhibition on the immune response. Furthermore, it addresses the clinical challenges faced in the clinic, including treatment resistance and the occurrence of irAEs.

## 2. Immune System and Immunotherapy

Immunotherapy represents a significant advancement in cancer treatment, whereby the host immune system is harnessed to eradicate tumor cells. This approach has transformed the field of oncology [5], as evidenced by its induction of a paradigm shift in how cancer is managed. Immune cells are the foundation for immunotherapy, as they play a pivotal role in regulating tumor progression [27,28]. The innate immune system works with the adaptive immune system, which activates specialized lymphocytes in a coordinated manner to fight against tumor cells (Figure 2A). Upon encountering a foreign antigen, naïve T cells establish a connection between their T-cell receptor (TCR) and the MHC, initiating a priming process. This initiates a series of signaling events that ultimately result in the activation of T cells [29,30]. However, additional co-stimulatory signals, such as the binding of Cluster of Differentiation 28 (CD28) to a B7 ligand, are necessary for the complete activation and proliferation of T cells [31] (Figure 2B). TCR/CD28 engagement activates several intracellular signaling pathways, which increase the production of cytokines, such as interleukin 2 (IL-2), further supporting T-cell expansion [32]. Although this process is essential for eradicating cancerous cells, inhibitory receptors, such as CTLA-4 and PD-1, are upregulated during T-cell activation to prevent excessive immune system stimulation. These receptors produce an inhibitory effect upon binding to specific B7 homologs on APCs or tumor cells. These inhibitory receptor–ligand interactions demonstrate the crucial role of ICs in fine-tuning the immune response, particularly in T-cell priming and regulating immune responses, including those within the tumor microenvironment (TME) [33]. They function as rheostats, adjusting the strength and duration of immune responses to align with the threat level and prevent potential damage to healthy tissues [34]. For example, during T-cell priming, CTLA-4 functions in the lymph nodes, competing with the co-stimulatory molecule CD28 for binding to B7 on APCs, regulating the activation of T cells [35]. Subsequently, during the effector phase within peripheral tissues, PD-1 interacts with its ligands PD-L1 and PD-L2, reducing the effector functions of T cells and promoting immune tolerance. This process allows for the fine-tuning of the immune response [36].

Cancer cells can exploit immune-checkpoint pathways within the TME to evade immune detection and destruction. Tumors can evade immune detection and destruction by overexpressing ICMs such as PD-L1, effectively “switching off” attacking T cells and promoting immune evasion and tumor growth [34]. Therefore, it is essential to maintain a delicate equilibrium between stimulatory and inhibitory signals to regulate immune responses and eradicate cancerous cells [37,38]. This new understanding of the intricate regulatory functions of ICs has prompted the development of several immunotherapies designed to inhibit immunosuppressive signals and restore antitumor immune responses. It is noteworthy that ICIs, such as anti-CTLA-4 and anti-PD-1 monoclonal antibodies (mAbs), have demonstrated considerable efficacy in treating various types of cancer [24]. Various therapeutic modalities have been developed to activate the immune system and manage cancers. These approaches include ICIs, cytokine therapies, oncolytic viruses (OVs), vaccines, and adoptive cell therapy (ACT).

### 2.1. Immune Checkpoints

Inherent immune system constituents, ICs are responsible for maintaining immune tolerance. They achieve this by transmitting signals to T cells, limiting the potential for an exaggerated immune response that may cause inadvertent harm to healthy cells. CTLA-4 is a type 1 transmembrane glycoprotein belonging to the immunoglobulin superfamily. It plays a pivotal role in regulating T-cell activation. Following engagement of the T-cell receptor (TCR), expression of CTLA-4 increases rapidly and reaches a peak 2 to 3 days after activation [39,40]. It inhibits TCR signaling by competing with CD28, a co-stimulatory molecule, for the B7 ligands B7-1 (CD80) and B7-2 (CD86), effectively attenuating T-cell activation [41,42,43] (Figure 3A). Most CTLA-4 is within intracellular vesicles and endosomal compartments throughout the Golgi apparatus, [44,45,46], and it is promptly transported to the immunologic synapse upon TCR engagement [47]. Upon TCR stimulation, kinases such as lymphocyte-specific protein tyrosine kinase (Lck) and ζ-chain-associated protein kinase 70 (ZAP-70) phosphorylate the cytoplasmic tail of CTLA-4 at tyrosine 165 (Y165). This phosphorylation event disrupts the interaction between CTLA-4 and adipocyte protein 2 (AP-2), enabling the retention of CTLA-4 on the cell surface within the immune synapse. This impedes T-cell activation by activating protein phosphatase 2A (PP2A), which inhibits AKT signaling [48]. When the intensity of the TCR signal is high, more CTLA-4 accumulates in the immune synapse and effectively outcompetes CD28 [49], providing a dynamic and adjustable inhibitory signal [47,50]. Moreover, CTLA-4 elicits inhibitory signals through its interaction with CD80/CD86 in APCs, which results in upregulation of IDO. This enzymatic activity results in localized tryptophan depletion, which inhibits effector T-cell function and promotes the activity of Tregs [51,52] (Figure 3A). IDO1 exerts an immunosuppressive effect by reducing CD8^+^ cell activity by converting tryptophan to kynurenine [53]. Another mechanism by which CTLA-4 exerts its suppressive effects is via a process called trogocytosis, whereby one cell takes up components from another. Upon binding to its ligands (B7 molecules) on APCs, CTLA-4, expressed on the surface of T cells, engages in trogocytosis [54]. This action effectively reduces the availability of B7 molecules for interaction with the co-stimulatory receptor CD28, thereby limiting T-cell activation and promoting immune tolerance [55]. Trogocytosis modulates the immune response, providing an additional layer of complexity to the immune regulation facilitated by CTLA-4.

The PD-1 receptor plays a pivotal role in regulating T-cell responses through programmed cell death signaling. It is also designated CD279 [56] and is expressed on many immune cells within the TME, including dendritic cells (DCs), activated monocytes, natural killer (NK) cells, T cells, and B cells [57]. PD-1 binds to two ligands, PD-L1 (B7-H1) and PD-L2 (B7-DC) [58], which are not only expressed in APCs and tumor cells but also in other non-hematopoietic cells [56,59] (Figure 3B). Upon engagement with its ligands, PD-1 suppresses T-cell immune responses [60] by recruiting the phosphatases Src homology 2 domain-containing protein tyrosine phosphatase 1 (SHP-1) and SHP-2, which dephosphorylate proximal TCR signaling molecules and inhibit T-cell activation [61,62,63]. Moreover, PD-1 inhibits T-cell activation by promoting the activation of SHP-2, which suppresses Akt signaling [63,64] (Figure 3B).

#### 2.1.1. Immune Checkpoint Inhibitors (ICIs) and Mechanism of Action

Unfortunately, cancer cells exploit checkpoints to evade immune detection and evade destruction by T cells [24,65,66]. Nevertheless, specific checkpoint blockade therapies have been demonstrated to effectively liberate the immune system from its constraints and facilitate the restoration of antitumor T-cell responses [4]. Monoclonal antibodies (mAbs), which target specific antigens, are multiplied thousands of times to produce a clinically effective therapeutic dose [67]. In 1997, rituximab was the first monoclonal antibody (mAb) approved by the Food and Drug Administration (FDA) to treat cancer, specifically non-Hodgkin’s lymphoma [68,69]. At present, immune-checkpoint inhibitors (ICIs) represent the most promising class of monoclonal antibodies (mAbs) in cancer research, with numerous drugs approved by the Food and Drug Administration (FDA) to treat over nine different cancer types [67]. This novel class of mAbs has become the cornerstone of immunotherapy and has significantly influenced cancer treatment [70]. The FDA has approved several ICIs, including ipilimumab, which targets CTLA-4, and nivolumab, Pembrolizumab, and Cemiplimab, which target PD-1. In addition, Atezolizumab, Durvalumab, and Avelumab, which target PD-L1, have also been approved. ICIs have shown considerable promise in treating diverse cancer types [71] (Table 1).

Ongoing clinical trials are investigating the expanding landscape of novel IC targets for the therapeutic blockade, including lymphocyte-activation gene 3 (LAG3), T-cell immunoglobulin (TIGIT), T-cell immunoglobulin domain, and mucin-domain 3 (TIM3). CD276 (B7-H3), natural killer group 2A (NKG2A), CD47, B and T lymphocyte attenuator (BTLA), CD200, indoleamine 2,3-dioxygenase 1 (IDO-1), and CD73 are being tested in clinical trials as potential inhibitors of these checkpoints [72] (Figure 4). These immunotherapeutic agents have transformed cancer treatment by reinforcing the host immune system to combat tumor cells [73].

ICI sales will likely reach up to USD 56.53 billion globally by 2025, representing the most rapid growth observed in the cancer treatment market. Several ICIs are presently being tested in clinical trials. However, variable therapeutic effectiveness and poor response have presented significant challenges [74]. These considerations warrant cautious optimism because of the tenuous link between initial positive results and therapeutic benefits. Such adverse events (AEs) are clinically substantial and may be classified as immune-related adverse events (irAEs) or toxicities that are clinically substantial [75]. Regarding tumor control, none of the recently developed ICIs appear to be as promising as PD-1/PD-L1 axis inhibition, at least when used alone. Consequently, many innovative compounds are being assessed with anti-PD-1 antibodies to overcome resistance to PD-1/PD-L1 blockade and provide a synergistic benefit that will enhance antitumor immunity. Moreover, given that immune checkpoints represent only a fraction of the intricate landscape of the immune system and TME, targeting them may be inadequate in numerous instances [72]. Nevertheless, combining ICIs with other immunotherapy techniques, such as radiation, chemotherapy, or targeted therapies, may be viable under certain circumstances.

ICIs specifically target immune checkpoint proteins, including CTLA-4, PD-1, and PD-L1. Typically, these proteins interact with their native ligands to transmit signals. For example, PD-1 typically binds to its ligands, PD-L1 and PD-L2, whereas CTLA-4 interacts with CD80 and CD86 to suppress T-cell activation. ICIs compete with these naturally occurring ligands to prevent inhibitory signals and maintain T-cell activity for binding to the immunological checkpoint proteins [76]. Steric hindrance refers to the process by which ICIs physically obstruct the binding sites of immunological checkpoint molecules, preventing natural ligands from interacting and enhancing the immune response against tumor cells [77]. Although ICIs rarely induce cell uptake directly, their binding to immune checkpoints can cause the internalization and degradation of these checkpoint proteins. This process, known as antibody-mediated endocytosis, reduces immune checkpoint molecules on the cell surface, further enhancing the immune response [78]. Moreover, the blockade of immune checkpoints by ICIs affects many intracellular signaling pathways. One of the primary pathways involves Rab GTPases, particularly Rab11, which control the trafficking of vesicles from recycling endosomes to the cell surface [79]. ADP-ribosylation factor 6 (Arf6), a small GTPase, plays a pivotal role in the formation and fusion of endosomal vesicles with the plasma membrane, which is facilitated by regulating actin cytoskeleton dynamics. The soluble N-ethylmaleimide-sensitive factor attachment protein receptor (SNARE) complex is also important for ensuring recycled immune checkpoint molecules are properly inserted into the plasma membrane [80,81].

Another crucial regulator is the endosomal sorting complex required for transport (ESCRT) machinery, which sorts membrane proteins into vesicles that bud from the endosomal membrane, directing immune-checkpoint molecules into recycling endosomes instead of lysosomes for degradation [82]. The phosphoinositide 3-kinase (PI3K)/AKT signaling pathway also influences the endocytic and recycling pathways, promoting the return of checkpoint molecules to the cell surface. Finally, protein kinase A (PKA) regulates endosomal recycling by phosphorylating various proteins involved in the trafficking processes, enhancing the recycling of immune-checkpoint molecules to the plasma membrane [83]. These interconnected pathways collectively ensure the efficient recycling of immune-checkpoint molecules, maintaining their ability to modulate immune responses as needed.

Recent developments in proteolysis-targeting chimera (PROTAC) technology have provided a novel approach for targeting immune checkpoints. PROTACs are bifunctional molecules that combine an E3 ubiquitin ligase and a target protein, promoting ubiquitination and the subsequent degradation of the target protein. In the context of immune checkpoints, PROTACs have the potential to selectively degrade proteins such as PD-1, PD-L1, and CTLA-4, which could provide a more efficient mechanism than traditional ICIs by permanently removing these proteins from the cell surface. This could cause a reduction in adverse effects and a decreased probability of developing resistance, which would represent a promising advancement in the field of current immune-checkpoint blockade therapies [84].

#### 2.1.2. Preclinical and Clinical Studies

The blockade of CTLA-4 induces tumor rejection through various mechanisms, as illustrated in Figure 5A. Structural analyses were conducted using crystallography of Ipilimumab: The CTLA-4 complex demonstrated that ipilimumab exerts its effects by inhibiting interactions with B7, representing its primary mode of action [85]. This inhibition primarily occurs within tumor-draining lymph nodes, where tumor antigens, including neoantigens and tumor-associated antigens (TAAs), can be efficiently cross-presented by antigen-presenting cells (APCs) to prime tumor-specific T cells. The absence of CTLA-4 has been demonstrated to reduce the threshold for TCR ligation, increasing the presence of active tumor-reactive T cells [49]. Ipilimumab, a human IgG1 monoclonal antibody, has been approved to treat melanoma and is being examined as monotherapy for other cancer types, including renal cell carcinoma (RCC), non-small cell lung carcinoma (NSCLC), and prostate cancer (PCa) [86]. A pooled meta-analysis of 10 prospective and two retrospective studies demonstrated that the three-year survival rate for patients with advanced melanoma who received ipilimumab therapy was 22% [87]. In a phase 1/2 clinical trial in melanoma patients, tremelimumab (a fully human IgG2 mAb) demonstrated an objective response rate (ORR) comparable to that observed with standard chemotherapy [88]. However, when used as monotherapy, it cannot enhance patient survival rates and is being investigated for use in combination with other regimens [4].

Inhibition of the PD-1 pathway enhanced the efficacy of antitumor T cells (Figure 5B). PD-1 blockade restores T-cell activity by regulating T-cell receptor (TCR) signaling and gene expression. Furthermore, blockade of PD-1 signaling has been demonstrated to partially reverse the metabolic alterations that lead to T-cell dysfunction [89]. Nivolumab is a fully human monoclonal antibody of the IgG4 subclass. This is an inaugural pharmaceutical agent approved as a PD-1 inhibitor. Nivolumab has been developed to selectively obstruct the interaction between the PD-1 receptor and its corresponding ligands, PD-L1 and PD-L2 [90]. In 2014 and 2015, Nivolumab was approved by the Food and Drug Administration (FDA) to treat melanoma and renal cell carcinoma (RCC), respectively [91]. Furthermore, it has also been approved to treat non-small cell lung cancer (NSCLC) [90], Hodgkin lymphoma [92], Head and Neck Squamous Cell Carcinoma (HNSCC) [93], hepatocellular carcinoma (HCC) [94], esophageal squamous cell carcinoma (ESCC) [95], pleural mesothelioma (PM) [96], and colorectal cancer (CRC) with mismatch repair deficiency (dMMR) or high microsatellite instability (MSI-H) [97]. Pembrolizumab (a humanized IgG4κ mAb) is an anti-PD-1 ICI that the FDA has approved to treat various types of cancers, including metastatic melanoma [98,99], NSCLC [98], HNSCC [100], and solid tumors with MSI-H [101]. Furthermore, it has been approved to treat advanced gastric cancer (GC) [102], cervical cancer (CC) [103,104], urothelial carcinoma (UC) [105], tumors with high mutational burden (TMB-H) [106], and triple-negative breast cancer (TNBC) [107]. Recently, the FDA approved the use of pembrolizumab to treat MSI-H or dMMR advanced endometrial carcinoma (EndC), as recommended by the KEYNOTE-158 trial [108]. Cemiplimab is another fully humanized IgG4 mAb approved by the FDA to treat cutaneous squamous cell carcinoma (CSCC) [109]. It has been demonstrated to have remarkable anticancer activity and a favorable safety profile [110]. In contrast to EGFR inhibitors and traditional chemotherapy, Cemiplimab has demonstrated superior overall survival (OS) and progression-free survival (PFS) [111].

Targeting PD-L1 with ICIs enhanced the activity of antitumor T cells (Figure 5B). The FDA approved three PD-L1 inhibitors: atezolizumab, durvalumab, and Avelumab. These monoclonal antibodies have applications in treating a variety of solid tumors, including NSCLC, head and neck squamous cell carcinoma (HNSCC), melanoma, and Merkel cell carcinoma (MCC) [112]. Atezolizumab is a humanized anti-PD-L1 mAb initially approved for UC [113], NSCLC [114,115], SCLC [116], melanoma [117], and HCC [118] due to its high response rate. Durvalumab, an additional anti-PD-L1 IgG1 mAb, was initially approved to treat urothelial bladder cancer (UBdC) [119] and subsequently approved for stage III and extensive stage NSCLCs. In 2022, the FDA approved durvalumab with chemotherapy for patients diagnosed with biliary tract cancer (BTC), as demonstrated in the TOPAZ-1 clinical trial [120]. Avelumab is a fully human IgG1 anti-PD-L1 monoclonal antibody that has also received FDA approval to treat metastatic MCC [121], UC [122], and RCC [123,124].

**Table 1 cancers-17-00880-t001:** List of FDA-approved Immune-Checkpoint Inhibitors.

Agent	ICI Target	FDA-Approved Indications	Ref.
Ipilimumab	CTLA-4	Metastatic melanoma	[125]
Nivolumab	PD-1	Metastatic melanoma	[91]
		Metastatic RCC	[91]
		Hodgkins’s lymphoma	[92]
		HNSCC	[93]
		HCC	[94]
		ESCC	[95]
		NSCLC	[126]
Pembrolizumab	PD-1	NSCLC	[98,99]
		Melanoma	[98,99]
		HNSCC	[100]
		GC	[102]
		CC	[103,104]
		UC	[105]
		TNBC	[107]
Cemiplimab	PD-1	Metastatic CSCC	[109]
Atezolizumab	PD-L1	UC	[113]
		Metastatic non-squamous and squamous NSCLC	[114,115]
		SCLC	[116]
		Melanoma	[117]
		HCC	[118]
Durvalumab	PD-L1	Locally advanced or metastatic urothelial carcinoma	[119]
		NSCLC	[127]
Avelumab	PD-L1	Merkel cell carcinoma MCC	[121]
		Urothelial carcinoma	[122]
		RCC	[123,124]

Abbreviations: RCC, Renal cell carcinoma; HNSCC, Head and Neck Squamous Cell Carcinoma; HCC, Hepatocellular carcinoma; ESCC, Esophageal squamous cell carcinoma; NSCLC, Non-small cell lung carcinoma; GC, Gastric carcinoma; CC, Cervical cancer; UC, Urothelial carcinoma; TNBC, Triple-negative breast cancer; SCLC, Small Cell Lung Cancer; MCC, Merkel cell carcinoma.

### 2.2. Cytokines and Cytokine-Based Therapies Against Cancers

The evolution of tumor immunology has marked a transformative era for cytokines, including tumor necrosis factor (TNF), interleukins (ILs), and interferons (IFNs). These cytokines have emerged as pivotal participants in the field of tumor immunology. Cytokines are small proteins released by immune and non-immune cells in response to various cellular stressors, including infection, inflammation, and tumorigenesis. They serve as signaling molecules facilitating cellular interactions and communication within the immune system [128]. Interferon alpha (IFN-α), the first cytokine to be discovered, is a prototypical therapeutic cytokine in cancer treatment [129]. Clinical studies have demonstrated that high doses of this cytokine have a therapeutic role in treating chronic myelogenous leukemia (CML) [130]. Given the pivotal role of IFN-α in T-cell priming [131], scientists have devised an innovative fusion protein that combines anti-PD-L1 and IFN-α. This fusion protein aims to facilitate targeted delivery of IFN-α directly to tumor sites, capitalizing on the heightened immunogenicity triggered by radiotherapy. The objective of this approach is to elicit a robust immune response. As evidenced by preclinical investigations, this strategy resulted in complete regression of tumors in most mice with B -cell lymphoma and CRCs following systemic administration of these fusion proteins [132]. Interleukin-2 (IL-2) was first identified in 1976 and has since been demonstrated to effectively enhance the production of T lymphocytes in large quantities when administered to individuals with metastatic cancer [133]. Consequently, it is frequently regarded as an immunosuppressive and “immunostimulatory cytokine” [67,134]. The FDA has acknowledged the potential of IL-2 as an immunotherapeutic agent to treat metastatic kidney cancer (KC) [67] and metastatic melanoma [135]. Despite its therapeutic potential, several intrinsic challenges have hindered the clinical application of IL-2. A significant concern pertains to the elevated toxicity of IL-2 relative to its therapeutic index, which imposes constraints on dosage and duration of administration. Furthermore, IL-2 has a relatively short half-life in the circulatory system, necessitating its frequent administration to maintain sustained therapeutic effects. The complex interactions of IL-2 with the immune system present a multifaceted challenge. It can promote immunosuppression via regulatory T cells (Tregs) and activate the immune system via other CD4^+^, CD8^+^, T, and natural killer (NK) cells. The outcome of this phenomenon is contingent upon the dose of IL-2 administered, with high doses stimulating the immune system and low doses having the opposite effect [136]. Several cytokines that act as integral mediators of the immune response have recently been the subject of human clinical trials because of their potential to treat various diseases [137]. However, the broader application of cytokines as therapeutic agents has been constrained by their unfavorable pharmacological properties, pronounced systemic effects, and pleiotropic behavior, which results in considerable toxicity and diminished efficacy [138]. Considering these challenges, there has been a notable increase in the development of innovative approaches that focus on cytokine engineering. These techniques are designed to enhance the therapeutic potential of cytokines by confining their activity to specific cells or tissues of interest and initiating their activity on demand. This approach improves targeted delivery of cytokines and minimizes systemic side effects [139]. A noteworthy advancement in this field is the recent approval by the FDA of a Biologics License Application (BLA) for N-803 (Anktiva™), an IL-15 agonist developed by ImmunityBio. The target population for this treatment comprises individuals diagnosed with non-muscle-invasive bladder cancer (NMIBdC) in situ who have not responded to standard Bacillus Calmette–Guérin (BCG) therapy [140,141,142]. Nevertheless, current research endeavors are exploring the possibility of utilizing these treatments combined with other immunotherapies, such as adoptive cell transfer (ACT) therapy, to effectively devise strategies to address the challenges posed by various diseases.

### 2.3. Oncolytic Virus-Based Therapies and Vaccines Against Cancers

OV therapy (OVT) is a revolutionary approach to cancer treatment that uses genetically modified viruses to target and eradicate cancer cells. Viruses have been genetically modified to reduce their virulence, enabling them to penetrate and lyse cancer cells [67]. In 2015, the FDA approved the first oncolytic virus therapy (OVT), Talimogene laherparepvec (T-VEC), as a treatment option for melanoma [143]. Oncorine (H101) is an oncolytic adenovirus therapy approved for cancer treatment by the US Food and Drug Administration (FDA). The China Food and Drug Administration (CFDA) approved this protocol in 2005. Oncorine is a genetically modified adenovirus designed to selectively replicate in and destroy tumor cells while sparing normal cells [144]. Another OV therapy, G47Δ (DELYTACT), was approved in Japan in June 2021. The Japanese Ministry of Health, Labour, and Welfare (MHLW) has granted conditional and time-limited approval to treat malignant glioma. This approval signified a noteworthy achievement, as DELYTACT became one of Japan’s inaugural oncolytic viral therapies sanctioned for cancer treatment. Other OVs, including Pexa-Vec (for HCC), CG0070 (for BdC), and G47Δ (for glioblastoma and PCa), have demonstrated promising outcomes in clinical trials [145]. Despite their success, a significant drawback of OVs is the development of acquired immunity specific to the virus, which may preclude repeat therapy in the same patient [145]. Among these combination therapies, a significant focus is on the concurrent use of oncolytic viruses (OVs) alongside other immunotherapies, particularly immune-checkpoint inhibitors (ICIs) and chimeric antigen receptor-engineered T-cell (CAR-T cell) immunotherapies. This symbiotic combination significantly enhanced cancer treatment regimens, as evidenced by studies [146,147]. Oncolytic viruses (OVs) have been observed to enhance the sensitivity of tumors to immune-checkpoint inhibitors (ICIs), a phenomenon often called the “heat-up” of tumors. This phenomenon, termed the “heat-up” of the TME, indicates that OVs can modify this environment to render it more conducive to an immune response, enhancing the efficacy of ICIs [148]. Moreover, OVs have demonstrated a distinctive ability to bolster CAR-T cells to evade the immunosuppressive TME (ISTME). This is achieved by effectively “piggybacking” oncolytic viruses on the CAR-T cells, enabling them to navigate the TME and enhance their antitumor efficacy against solid tumors [149]. The dual functionality of OVs highlights their potential to revolutionize the landscape of cancer therapy.

In 1990, the Food and Drug Administration (FDA) approved the Bacillus Calmette–Guérin (BCG) vaccine to treat superficial bladder cancer, marking the first instance of a cancer vaccine receiving regulatory approval. Subsequently, the sipuleucel-T vaccine was approved by the Food and Drug Administration (FDA) to treat castration-resistant prostate cancer (PCa) to extend the overall survival (OS) of patients [150]. Notwithstanding these advances, the efficacy of cancer vaccines has been constrained by insufficient comprehension of the means to elicit a cytotoxic T-cell response in patients and to surmount the TME to achieve an antitumor response that results in clinically meaningful tumor death. It is imperative to address these challenges to enhance the efficacy of cancer vaccines [34].

### 2.4. Adoptive Cell Therapy (ACT)

ACT represents a novel approach to cancer treatment that leverages the patient’s immune system to combat tumors. This treatment entails either isolating and reintroducing tumor-infiltrating lymphocytes (TILs) that are inherently reactive to the patient’s tumor or genetically modifying T cells to express transgenic T-cell receptors (TCRs) or chimeric antigen receptors (CARs) that target specific tumor antigens. In this therapeutic approach, autologous immune cells are isolated, genetically modified as required, expanded ex vivo, and reintroduced into the patient to achieve persistent clinical efficacy [151,152,153]. In contrast, allogeneic ACT significantly differs from the conventional approach of utilizing immune cells derived from a patient’s cellular repertoire. Instead, it involves the use of immune cells sourced from a healthy donor, a strategy with the potential to broaden the scope of available cellular therapies [154]. Complex modifications are frequently used to render donor-derived cells suitable for therapeutic applications. These modifications are paramount in mitigating the significant concern of graft-versus-host disease (GVHD), a severe and potentially life-threatening condition characterized by donor-derived immune cells mounting an immune attack on the recipient’s tissues [155]. Moreover, these modifications have been meticulously devised to mitigate the risk to the patient’s immune system by inducing rejection of infused allogeneic cells [156]. One advantage of allogeneic ACT over its autologous counterparts is its economic feasibility and scalability [157]. Using donor cells facilitates the creation of donor cells and “off-the-shelf” therapeutic solutions. Such standardized therapies can be mass-produced, stockpiled, and administered to multiple patients without requiring individualized tailoring. Consequently, allogeneic ACT has the potential to extend therapeutic reach. It has the potential to significantly reduce economic burden and streamline treatment processes, ushering in a new era of equitable access to advanced immunotherapeutic approaches. One of the most significant advancements in ACT is the emergence of CAR-T cell therapy. Despite the occasional occurrence of severe adverse effects, such as cytokine release syndrome, which necessitates immediate therapeutic intervention, CAR-T-cell therapy has demonstrated its efficacy in cancer treatment [158,159]. Moreover, premanufactured models have facilitated the standardization of CAR-T cell products, ensuring consistent quality and potency across different batches of CAR-T cells. Furthermore, the extended preparation window before patient administration allows researchers to introduce multiple cellular modifications, facilitating redosing strategies and developing therapeutic combinations that target various cellular antigens. From an economic standpoint, the allogeneic model offers scalability and cost-effectiveness through an industrialized process, making it a viable option for enhancing the accessibility of cellular immunotherapies to a broader patient population [160]. The advent of CAR-T-cell therapy has marked a revolutionary shift in the treatment landscape of hematological malignancies, with unprecedented response rates observed, particularly in non-Hodgkin lymphoma (NHL), relapsed/refractory (R/R) B-cell acute lymphoblastic leukemia (B-ALL), and multiple myeloma (MM) [161]. The US Food and Drug Administration (FDA) has approved six CAR-T-cell therapies to treat hematological cancers [162]. The list is led by Kymriah (tisagenlecleucel) and Yescarta (axicabtagene ciloleucel), the pioneering CAR-T therapies approved to treat patients with relapsed/refractory B-cell precursor acute lymphoblastic leukemia (R/R B-cell ALL) and relapsed/refractory diffuse large B-cell lymphoma (R/R DLBCL), respectively [163,164,165]. The FDA-approved CAR-T cells primarily target CD19 and B-cell maturation antigen (BCMA) [163,165,166,167,168,169]. Allogeneic CAR-T cells derived from donor sources have led to substantial advancements in cellular immunotherapy. In contrast to autologous approaches, the potential advantages inherent in utilizing allogeneic CAR-T cells have propelled this approach to the forefront of contemporary immunotherapy research and development. Techniques involving TILs, genetically engineered T cells expressing advanced T-cell receptors (TCR), or CAR represent innovative approaches to effectively stimulate the immune system. The primary objective is to enhance the ability of the immune system to effectively recognize and eliminate malignant cells [170]. It is noteworthy that patients who exhibit positive responses to ICI therapy often have tumors characterized by T-cell inflammation. This highlights the necessity for the development of strategies to transform non-T-cell-inflamed TMEs into inflamed environments, rendering them more susceptible to immune-mediated eradication. Recent studies have yielded promising results. For example, the combination of anti-PD-L1 antibodies with TILs has demonstrated the potential to enhance T-cell infiltration and increase the production of IFN-gamma (IFN-γ) in tumor-bearing mice, resulting in a reduction in tumor growth [171]. Clinical trials combining TILs with established immunotherapy drugs such as ipilimumab or nivolumab in patients with ovarian cancer (OVC) have yielded promising results, including partial response and disease stabilization [172]. Moreover, groundbreaking research has revealed that CAR-T cells genetically modified to secrete PD-1-blocking single-chain variable fragments (scFv) can enhance antitumor immunity in vivo [173]. In a related therapeutic development, combining a PD-1 inhibitor with mesothelin-specific CAR-T cells has been explored in patients with malignant pleural mesothelioma (MPM) [174,175]. These studies have demonstrated that innovative approaches to cancer immunotherapy are safe and, sometimes, moderately effective, underscoring their potential for further development.

### 2.5. Bispecific Immune-Checkpoint Blockade Antibodies

Bispecific IC blockade antibodies represent an innovative approach in cancer immunotherapy, offering a promising strategy to enhance the effectiveness of ICIs by simultaneously targeting multiple ICs. Engineered to bind to two different targets, typically by pairing an inhibitory receptor on T cells with a tumor-associated antigen, these antibodies are designed to activate T cells and induce tumor cell destruction [176]. The rationale behind this approach is to overcome the limitations of single-agent immunotherapy, such as the development of resistance mechanisms and the heterogeneous nature of the TME.

New knowledge regarding the role of immune checkpoints in enabling cancer cells to evade the innate and adaptive immune systems has brought about a revolutionary transformation in clinical cancer therapy. Immune-checkpoint receptors, such as PD-1 and cytotoxic CTLA-4, are vital for maintaining self-tolerance and preventing immune-mediated damage to the host. Nevertheless, evidence indicates that TILs frequently exhibit markedly elevated levels of these co-inhibitory receptors, resulting in an exhausted phenotype with diminished antitumor activity [124]. Furthermore, preclinical evidence indicates that elevated PD-L1 expression in malignant cells impedes T-cell responses, facilitating tumor immune escape and reducing the efficacy of antitumor immunotherapies. To counteract these mechanisms, immune-checkpoint blockade (ICB) has been developed to disrupt these negative regulators and activate pre-existing antitumor immune responses. Some ICBs have demonstrated remarkable efficacy against various cancers and have been incorporated into standard clinical practice. For example, a clinical trial reported five-year outcomes indicating that the combination of nivolumab (anti-PD-1) and ipilimumab (anti-CTLA-4) in patients with advanced melanoma resulted in sustained long-term progression-free and overall survival rates of 52% compared to 44% in the nivolumab group and 26% in the ipilimumab group [177].

Simultaneous blocking of two inhibitory immune-checkpoint (IC) molecules has driven the development of bispecific antibodies (BsAbs) that can target two checkpoints in the same or different cells. This innovation has its roots in the success of ICB therapies and the improved clinical outcomes observed in patients treated with combined ICBs [176]. MGD019 is a monovalent, investigational PD-1 × CTLA-4 bispecific DART compound designed to enhance CTLA-4 blockade in the TME via PD-1 binding. This single molecule effectively blocked the PD-1/PD-L1 axis and variable inhibition of CTLA-4 in vitro, exhibiting favorable tolerability in nonhuman primates with augmented T-cell proliferation and expansion [178]. An ongoing first-in-human study on MGD019 in patients with multiple advanced solid tumors demonstrated acceptable safety and objective responses in various tumor types that were typically unresponsive to ICIs (NCT03761017) [178].

MEDI5752 is a bispecific antibody that combines an anti-PD-1 monoclonal antibody with the variable binding domains of tremelimumab (anti-CTLA-4) on the DuetMab backbone. It has been engineered with triple amino acid mutations in the human IgG1 constant heavy chain, which reduces Fc-mediated immune effector functions. Research by Dovedi, S.J. et al. showed that this molecule selectively targets and inhibits CTLA-4 in PD-1-positive T cells, leading to the internalization and degradation of PD-1. This results in an enhancement of the affinity and saturation of CTLA-4 receptors, increasing their clinical benefits and reducing potential harm. Initial clinical trials of MEDI5752 in patients with advanced solid tumors have yielded encouraging results, with a notable proportion of patients exhibiting partial responses while experiencing manageable adverse effects [179]. In contrast to MEDI5752, AK104 is an anti-PD-1/CTLA-4 bispecific antibody with a symmetrical 4-valent IgG1-scFv structure developed by Akeso Biology. AK104 facilitates the independent endocytosis of PD-1 or CTLA-4, demonstrating good antigenic differentiation and high retention in the tumor tissue. Recently, Cadonilimab (AK104) was approved in China to treat recurrent or metastatic cervical cancer in patients who have failed prior platinum-containing chemotherapy [180].

Most bispecific antibodies (BsAbs) target the subsequent inhibitory receptors on tumor-infiltrating lymphocytes (TILs) with one binding arm while simultaneously blocking the programmed death-1 (PD-1)/PD-ligand 1 (PD-L1) axis with the other to reverse T-cell exhaustion-driven resistance. For example, MGD013 targets LAG-3 and PD-1, which are expressed on T cells following antigen stimulation. Based on the DART^®^ platform, MGD013 has been demonstrated to effectively inhibit the binding of PD-1 to PD-L1 and PD-L2 and the binding of LAG-3 to MHC II. This results in the activation of T cells. This bispecific antibody has been evaluated in a phase I clinical trial (NCT03219268). Additional preclinical dual immunomodulators include FS118, which targets PD-L1/LAG-3, and LY3415244, which targets PD-L1/ T cell immunoglobulin and mucin domain-containing protein 3 (TIM3) [181].

## 3. Combination of Immunotherapies

The efficacy of T-cell activation and the ensuing immune response hinge on a fine equilibrium between co-stimulatory and co-inhibitory signals. To circumvent resistance to PD-1 and CTLA-4 blockade, considerable research has been directed toward targeting additional co-inhibitory immune checkpoint receptors. It has been shown that these receptors are frequently upregulated in regulatory T cells (Tregs) and are associated with T-cell exhaustion [182]. Strategies to mitigate resistance to anti-PD-1/PD-L1 therapies include targeting these alternative IC receptors or enhancing co-stimulatory signaling [53]. This approach has been supported by numerous preclinical studies that have demonstrated that the inhibition of various checkpoint receptors, such as TIM-3 [183] and TIGIT [184], in conjunction with anti-PD-1 therapy, can significantly improve patient outcomes. Several clinical trials have evaluated the efficacy of antibodies targeting a range of IC receptors, including [184,185,186]. This comprehensive approach highlights the intricate nature of immune regulation in cancer therapy and the possibility of targeted interventions to enhance immunotherapy efficacy. T-cell activation can be enhanced through the use of co-stimulatory agonists, including TNF ligand superfamily member 9 (4-1BB), OX40, CD40, glucocorticoid-induced TNFR-related protein (GITR), and inducible T-cell co-stimulator (ICOS). The combination of these co-stimulatory molecules with ICIs represents a compelling basis for further investigation in the context of cancer therapy. Although T-cell exhaustion is frequently regarded as irreversible, the judicious deployment of co-stimulatory agonists is designed to bolster cytotoxic T lymphocyte (CTL) proliferation, survival, and effector function [187]. This approach optimizes the overall immune response, potentially overcoming some of the limitations associated with monotherapy treatments.

Monotherapies have made significant advances in cancer treatment. Nevertheless, further improvements must enhance their efficacy across various tumor types. Researchers are investigating the potential of combinatorial immunotherapy regimens involving ICIs that regulate T-cell function via separate mechanisms to address this challenge. Clinical trials have demonstrated that the combination of ICIs, such as nivolumab and ipilimumab, enhances progression-free survival (PFS) in cancer patients, particularly in those with melanoma [188]. In 2020, the FDA approved the use of nivolumab plus ipilimumab as first-line therapy for unresectable malignant pleural mesothelioma (MPM) and non-small cell lung cancer (NSCLC) [189]. This combination has been approved for various types of cancers, including solid and liquid tumors [60], and has demonstrated efficacy in advanced RCC, MSI-H/dMMR metastatic CRC, and HCC [2]. The KEYNOTE-021 trial (NCT02039674) investigated the efficacy of combining Pembrolizumab and Ipilimumab as second-line therapy for patients with stage IIIB/IV NSCLC. This combined approach yielded notable outcomes, including partial and complete responses, in certain patients [190]. Notwithstanding the success of anti-PD-1/PD-L1 and anti-CTLA-4 and their combination, efforts are being made to enhance their efficacy and mitigate toxicity using ICIs. Given the RELATIVITY-047 clinical trial findings, the FDA recently approved using relatlimab, a medication targeting lymphocyte-activation gene 3 (LAG-3), with nivolumab to treat unresectable or metastatic melanoma. Several recent trials have demonstrated that combining ICIs with CT or RT provides a significant survival advantage [191,192]. Radiotherapy and chemotherapy increase cytosolic DNA, cause DNA damage, and induce neoantigens, prompting the host immune response [193]. These damaged immune cells contribute to developing sub-clonal mutations associated with the tumor’s capacity to evade immune detection and mount a poor response to ICIs [194]. Consequently, scientific and clinical investigations of ICIs combined with targeted therapies are becoming increasingly important.

Dysregulation of the DNA damage response (DDR) plays a significant role in developing genomic instability [195,196]. In contrast, tumor DNA damage response (DDR) gene alterations are strongly associated with tumor susceptibility to immune-checkpoint inhibitors (ICIs) [197,198]. The accumulation of damaged DNA subsequently elicits an antitumor immune response [199]. Furthermore, relocating damaged DNA from the nucleus to the cytoplasm is essential for activating IFN genes via the STING/TBK1/IRF3 pathway. This process initiates an innate immune response that is intricately linked to a consistent and long-lasting reaction to ICIs [200,201]. An expanding body of research has demonstrated that targeting the DNA damage response (DDR) effectively promotes inflammation in the tumor immune microenvironment (TIME). This approach has enhanced immune recognition and malignant cell death, particularly when combined with immune-checkpoint inhibitors (ICIs) [202,203]. An increasing number of ongoing clinical trials are investigating novel agents that target DNA damage response (DDR) pathways combined with ICIs [199]. These agents include cyclin-dependent kinase 4/6 inhibitors (CDK4/6i), poly (ADP-ribose) polymerase inhibitors (PARPi), ataxia telangiectasia inhibitors, and Rad3-related (ATR) kinase inhibitors, WEE1 inhibitors, checkpoint kinase 1 (CHK1) inhibitors, and DNA-dependent protein kinase (DNA-PK) inhibitors. Among these, the combination of PARP inhibitors with PD-1/PD-L1 inhibitors is the furthest in clinical development [199]. For example, the combination of niraparib (PARPi) and pembrolizumab has been tested in 122 patients with advanced or metastatic (TNBC or recurrent OVC) disease in a multicenter, open-label, phase 1/2 trial to determine its effectiveness and safety. The combination of niraparib and pembrolizumab has a tolerable safety profile and shows promising antitumor efficacy [204,205]. Preliminary data suggest that the combination of drugs targeting DNA damage repair with immune-checkpoint inhibitors represents a promising strategy for cancer treatment.

## 4. Immune-Related Adverse Events (IrAEs)

Immune-related adverse events (IrAEs) are immunological events in which the heightened immune activity erroneously identifies normal host tissues as foreign invaders. These irAEs may manifest as dermatological, gastrointestinal, endocrine, or other organ-specific complications, reflecting the diversity of tissues affected by the immune response [206]. The precise pathophysiological mechanisms remain unclear; however, they are associated with alterations in the immune system. Such processes include a breakdown in tolerance and heightened sensitivity to antigen recognition, which may involve molecular mimicry [207]. Molecular mimicry refers to a phenomenon whereby the immune system erroneously targets both foreign and self-antigens, contributing to the development of irAEs [208]. The implicated mechanisms include excessive autoantibody production, increased T-cell infiltration into various organs, and the release of inflammatory cytokines. These cytokines further exacerbate the immune response, creating a detrimental cycle of immune-mediated damage [207].

Environmental factors, including disruptions in the equilibrium of the intestinal microbiota and the production of metabolites by the microbiota, have been demonstrated to induce aberrant activation of the immune system. This heightened immune activation represents the underlying etiology of irAEs during ICI therapy. These irAEs can manifest across several severities, from relatively minor dermatological conditions to more critical illnesses, such as myocarditis or colitis [209,210]. The most prevalent cutaneous irAEs are rash, vitiligo, Stevens–Johnson syndrome, and toxic epidermal necrosis. Conversely, gastrointestinal irAEs such as diarrhea and colitis are higher with anti-CTLA-4 agents [211,212]. Anti-PD-1 agents are more frequently associated with thyroid dysfunction [213,214], whereas hypophysitis is rare with anti-PD-1 agents but is common with anti-CTLA-4 agents [215,216]. Furthermore, ICIs have been demonstrated to induce autoimmune diseases, including type I diabetes, even in previously asymptomatic patients [210,217]. The incidence and severity of irAEs vary among checkpoint therapies, with anti-PD-1 demonstrating a superior safety profile compared to anti-CTLA-4. The combination of anti-CTLA-4 and anti-PD-1 antibodies has markedly elevated the risk of irAEs [188]. Most irAEs can be reversed with steroid administration if diagnosed and treated promptly [218]. Although endocrine disorders may be permanent, they can be effectively managed with hormone replacement therapy [218]. Although ICIs can provide long-term benefits and cures, patients should be monitored for several years to avoid late-onset irAEs [219]. The safety of ICIs remains unproven, and patients with underlying autoimmune disorders have been excluded from clinical trials [219]. Guidelines for managing irAEs have been published, and physicians will rule out other conditions before initiating treatment. It is of utmost importance that patients and their physicians engage in the frequent monitoring and management of irAEs. This process should include a discussion of the potential risks and benefits associated with these therapies [218]. Understanding the intricate immunological mechanisms underlying irAEs is essential to optimize cancer immunotherapy and manage these side effects, ensuring patient safety and treatment efficacy.

### 4.1. Autoreactive T Cells

Maintaining a delicate equilibrium between immune activation and tolerance is paramount in immune regulation, which is orchestrated through the co-stimulatory pathway of reactive T cells [220]. Immune tolerance is a vital mechanism for curbing the activation of self-reactive T cells, thereby modulating the overall strength of the immune system. Inhibitory co-stimulatory molecules on T cells play a pivotal role in maintaining the equilibrium between T-cell activation, immune tolerance, and the potential for immune-mediated tissue damage by engaging their ligands [221]. In this context, ICIs play a dual role. On one hand, they facilitate T-cell activation and proliferation, which can be advantageous in the fight against cancer. However, ICIs also impede the function of regulatory T cells (Tregs), which are responsible for maintaining immune tolerance. The number of regulatory T cells (Tregs) is inversely correlated with the incidence of immune-related adverse events (irAEs). Consequently, ICIs disrupt peripheral T-cell tolerance, resulting in heightened inflammation and autoimmunity, particularly in organs that rely heavily on peripheral T-cell tolerance mechanisms, such as the skin and colon [222,223,224]. Recent research has illuminated the correlation between severe irAEs and elevated levels of CD4 effector memory T (TEM) cells in the blood, indicating these cells may serve as the foundation for severe ICI toxicity [225].

Scientists have established a correlation between the substantial clonal diversity of TCRs within activated CD4^+^ TEM cells and individuals who experience severe irAEs. However, this correlation was weaker or absent in other T-cell subpopulations [226]. Favorable responses to ICIs are associated with higher proportions of T lymphocyte subsets such as CD4^+^ and CD8^+^ T cells [227], CD45RO^+^ CD8^+^ memory T cells, and Tregs [228,229] during treatment. These are associated with improved prognosis and potentially enhanced antitumor immune responses.

### 4.2. Autoreactive B Cells

The activation of self-reactive B cells results in an increased production of auto-Abs, which may originate from new clones or be derived from existing ones [230]. These autoantibodies have the potential to bind to their target antigens, which could cause harm through the initiation of the classical complement cascade reaction, antibody-dependent cellular cytotoxicity (ADCC), or the deposition of immune complexes [231]. One study found that 19.2% of patients with no detectable autoantibodies before ICI treatment developed autoantibodies following treatment. These findings underscore the pivotal role of B cells in regulating immunological response to ICI therapy. The presence of autoantibodies does not necessarily indicate the presence of autoimmune diseases or immune-related adverse events (irAEs), despite the correlation between the two. These emerging autoimmune antibodies predominantly target thyroid-related antigens, specifically thyroid peroxidase (TPOAb) and thyroglobulin (TGAb), as indicated in a previous study [232].

### 4.3. Cytokine Storm

In patients who experienced irAEs, notable alterations in the cytokine profiles were observed both before and after treatment. These cytokines have the potential to act as signaling molecules, enhance immune system responses, and contribute to the development of irAEs. Immune cells can cause tissue damage through various mechanisms when they release inflammatory mediators, particularly in anatomically sensitive areas. Such mechanisms include systemic reactions, such as fever, local immune-mediated injury, and compromised tissue perfusion. Local immune injury results in direct cytotoxic effects and the formation of inflammatory microenvironments that alter normal tissue function. Impaired perfusion can also lead to ischemia and necrosis. This finding suggests that specific or overall cytokine levels may be involved in the pathogenesis of irAEs [233]. Cytokines can engage immune cells and activate intracellular signaling pathways, including JAK-STAT and PI3K-AKT-mTOR. Such pathways may cause an imbalanced proinflammatory response, which could exacerbate irAEs.

Lower baseline levels of IL-6 have been linked to an elevated risk of developing irAEs [234]. Furthermore, the efficacy of TNF inhibitors in mitigating irAEs was associated with inflammatory factors. Microbiota, defined as a community of microorganisms residing in the body, has been demonstrated to substantially influence the development of irAEs by regulating the production of proinflammatory or anti-inflammatory cytokines [235]. This effect was more pronounced following the treatment with ICIs. As evidenced in preclinical murine models, IC inhibition can elicit particular inflammatory T-cell responses associated with the microbiota [236]. These unconventional responses involve symbiont-specific T cells that produce IL-17, which induces dermatological inflammation observed in patients undergoing treatment with ICIs.

## 5. Resistance to Immunotherapy

Despite the remarkable clinical efficacy of immune checkpoint inhibitors (ICIs) across a spectrum of malignancies, many patients fail to respond or develop resistance to these agents. Resistance can be classified as either primary or secondary. Primary resistance refers to the absence of an initial response, whereas secondary resistance refers to a relapse or progression following an initial response. The response rates to ICI treatment vary considerably across different cancer types. For instance, Hodgkin’s lymphoma shows high response rates, whereas MSI-H CRC often exhibits minimal to no response [197,237]. A deeper understanding of the mechanisms underlying resistance to ICIs is vital to increasing the number of patients who can benefit from this form of cancer immunotherapy. The resistance mechanisms discussed below have been characterized thus far, and ongoing research continues to uncover further mechanisms.

### 5.1. Primary Resistance

Primary resistance is defined as the inability of tumors to respond to initial immunotherapy treatment. This resistance is evident in clinical contexts, where patients do not exhibit a therapeutic response to specific treatments. For example, approximately 40–60% of patients with melanoma exhibit primary resistance to nivolumab treatment, whereas the resistance rate for ipilimumab is higher, with over 70% of patients demonstrating non-responsiveness [6,188,211,238,239]. A comprehensive understanding of primary ICI resistance mechanisms is essential to optimize therapeutic approaches and improve patient outcomes. The subsequent sections elaborate on these mechanisms, particularly as they manifest in immuno-oncology (Figure 6).

#### 5.1.1. Tumor Mutational Burden and Neoantigen Expression in Immunotherapy Response and Resistance Mechanisms

The capacity of tumors to elicit adaptive immune responses is contingent on the immune system, which identifies tumor cells as foreign entities. High TMB and elevated neoantigen expression are essential for promoting antitumor immunity [240,241]. Cancers with an elevated mutational burden have been observed to demonstrate a greater likelihood of responding favorably to anti-CTLA-4 and anti-PD-1/PD-L1 therapies across a range of cancer types [10,242,243,244,245]. Elevated TMB is not a reliable indicator of a tumor’s potential response to ICIs. However, a high neoantigen load increases tumor visibility in the immune system, leading to enhanced antitumor T-cell responses after ICI therapy [10]. It has been observed that tumors with high MB are more likely to generate immunogenic neoantigens capable of inducing tumor rejection, thus potentiating their response to ICI therapy in melanoma and NSCLC [10]. Conversely, low TMB and low availability of neoantigens represent critical mechanisms for primary resistance to ICI immunotherapy. This is evidenced by observations in PCa and PC with low MB, which are less likely to benefit from therapy owing to the absence of immunogenic neoantigens [246,247].

#### 5.1.2. Antigen Processing and Presentation Dysfunction and ICI Therapy Resistance

The antigen processing (AP) and presentation (APP) pathways are of paramount importance for a successful response to immune-checkpoint inhibitor (ICI) therapy [248]. This multistep process entails processing a peptide using cellular AP machinery, with subsequent loading of the peptide onto an MHCI molecule for presentation. The APP pathway comprises several essential components whose aberrant expression is linked to a reduction in tumor-infiltrating CD8^+^ T cells and a lack of responsiveness to ICI-based treatment [249,250,251]. Beta-2 macroglobulin (β2M) is essential for the stability of HLA-I molecules. Mutations in β2M can cause MHC instability, which can cause AP failure [252]. For example, the loss of β2M in NSCLC results in the absence of HLA-I molecule expression, which causes resistance to ICIs [250]. In addition, non-responders to PD-1 blockade in melanoma exhibit a higher loss of heterozygosity at the β2M locus [253]. Transporters associated with antigen processing (TAPs) play a pivotal role in translocating peptides for antigen presentation. Their downregulation has been observed in various cancers, reducing major histocompatibility complex (MHC) I surface expression and altering the antigenic peptide repertoire [20,254,255,256,257]. Tapasin plays a role in loading high-affinity peptides onto MHC I molecules; its altered expression has been observed in multiple cancers. This reduces MHC I surface expression and contributes to tumor progression [258,259,260]. Calreticulin plays a role in properly functioning antigen presentation and immune surveillance. In cancers such as colorectal cancer (CRC), breast cancer (BC), and myeloproliferative neoplasms, reduced expression of calreticulin has been linked to compromised APP [261,262,263]. These components are essential for effective immune surveillance and tumor eradication. Their disruption significantly affects the success of ICI therapy in various cancer types.

#### 5.1.3. Immunosuppressive Dynamics in the Tumor Microenvironment and ICI Therapy Resistance

The isTME in cancer cells comprises various components that contribute to resistance to ICIs [264]. Tumor metabolites, immunosuppressive cells, and cytokines can impede the anticancer activity of ICIs [265,266]. Tregs exert a detrimental effect on T-cell functionality, which impairs the function of CD8^+^ T cells [267]. Tregs generate immunosuppressive substances, including IL-10, extracellular adenosine, and transforming growth factor-beta (TGF-β), which suppress IL-2, exacerbate immune dysregulation (by impairing the proliferation and function of effector T cells), and promote immune escape [268,269]. Tumor-associated macrophages (TAMs), particularly those with the M2 phenotype [270,271], secrete inhibitory cytokines and additional suppressive elements that suppress CTLs, recruit immunosuppressive cells, and promote tumor progression [272]. These effects contribute to poor clinical outcomes. For example, TAMs contribute to immunosuppression in PC, resulting in resistance to PD-1/PD-L1 therapy. Inhibition of the colony-stimulating factor 1 receptor (CSF1R) on TAMs has been demonstrated to increase the expression of PD-L1 and CD8^+^ T-cell infiltration, eliminating anti-PD-1/PD-L1 resistance [273]. Furthermore, reprogramming of TAMs into the M1 phenotype restores ICI resistance [274]. Conversely, a shift from M2 to M1 macrophages has been demonstrated to enhance PD-L1 expression. Consequently, ICIs with TAM reprogramming may represent a pioneering therapeutic strategy [275] for tumor regression [272,276].

Myeloid-derived suppressor cells (MDSCs) are a heterogeneous population of bone marrow-derived cells that can suppress immune responses [10]. MDSCs have been demonstrated to facilitate tumor invasion, metastasis, and angiogenesis [277,278]. Several clinical studies have demonstrated that a poor response to ICIs is associated with an elevated number of MDSCs within the TME [279,280]. Several studies have confirmed a correlation between MDSC infiltration and resistance to anti-PD-1 therapy. There is evidence that targeted depletion of MDSCs can aid in restoring the effectiveness of anti-PD-1 therapy [281,282,283]. Therefore, the efficacy of anti-PD-1/PD-L1 treatment is undermined by MDSCs in the TME. Furthermore, research indicates that impeding the migration of MDSCs to the TME can enhance the efficacy of anti-PD-1 therapy and overcome resistance to PD-1 blockade [282,284,285]. The role of transforming growth factor-beta (TGF-β) in the TME is multifaceted and complex. It accelerates tumor growth by causing tumor cells to undergo epithelial–mesenchymal transition (EMT), attracting immunosuppressive cells, including MDSCs and Tregs, and impairing CD8^+^ T-cell activity [286]. In a murine CRC model, elevated TGF-β signaling was linked to poorly immunogenic tumors, which exhibited a limited response to ICI, suggesting a resistance state [287]. These findings are corroborated by evidence that TGF-β inhibition enhances the anticancer response to ICI in metastatic UC [288]. The cytokine milieu within the TME exerts both immunosuppressive and stimulatory effects [289]. Some chemokines, including CXCL8, CXCL12, CCL5, CCL22, and CCL17, have been demonstrated to promote an immunosuppressive environment by recruiting MDSCs and Tregs to the TME [282,290]. Inhibition of the chemokine receptor CCR4 has been shown to reduce the trafficking of regulatory T cells (Tregs) and enhance antitumor effects [291,292]. In contrast, CXCL9 and CXCL10 facilitate the recruitment of cytotoxic T lymphocytes (CTLs) to the TME, destroying cancer cells [293,294]. However, epigenetic mechanisms may impede chemokine expression. The subsequent reduction in TILs resulting from their silencing can confer resistance to ICIs, impeding their overall therapeutic potential. In OVC, epigenetic suppression of CXCL9 and CXCL10 has been observed [295]. A previous study demonstrated that treatment with an epigenetic modulator could effectively counteract this suppression. Specifically, treatment with this modulator restores the expression of these critical chemokines, enhancing the response to ICIs [295]. In addition to chemokine modulation, the metabolic landscape of the TME is of great consequence to immune responses. The accumulation of metabolites such as kynurenines and the subsequent reduction in the essential amino acid tryptophan have been demonstrated to have notable immunosuppressive effects. This imbalance facilitates immune evasion through mechanisms that induce T-cell anergy and apoptosis, exacerbating the challenge of effective immunotherapy [296].

#### 5.1.4. T-Cell Exhaustion and Resistance to Immune-Checkpoint Blockade

When T cells are exposed to prolonged tumor antigens and an immunosuppressive TME, they may undergo a “T-cell exhaustion”. This state is distinguished by the sustained expression of inhibitory receptors and a transcriptional profile that diverges from that of functional memory or effector T cells [297]. Despite the potential for interventions, such as PD-1/PD-L1 blockade, to rejuvenate specific exhausted T cells and restore immune function, some T cells ultimately enter an irreversible exhaustion state. This irreversible state is postulated to underlie the resistance to anti-PD-1/PD-L1 therapies [298,299]. Furthermore, the degree of T-cell surface PD-1 expression influences the efficacy of anti-PD-1 therapy [300,301]. CD8^+^ T cells exhibiting high levels of exhaustion and PD-1 expression do not respond to PD-1 blockade. However, T cells expressing PD-1 at intermediate levels are responsive. While considerable attention has been directed toward the role of PD-1 and PD-L1 interactions, it is important to recognize that other co-inhibitory molecules expressed by cancer cells can also contribute to resistance [299]. In addition to PD-1, PD-L1, and CTLA-4, recent findings have underscored the importance of T-cell immunoglobulin and mucin-domain 3 (TIM-3), LAG-3, and T-cell immunoreceptors with Ig and ITIM domains (TIGIT) in immunology. TIM-3 is expressed in various immune cells, including regulatory T cells (Tregs), natural killer (NK) cells, DCs, and monocytes, to prevent excessive immune activation [182]. In studies focusing on cancers such as melanoma and NSCLC, a notable finding is the overexpression of TIM-3 in Treg cells, which has been implicated in developing resistance to anti-PD-1 therapies [183,302]. This resistance mechanism was partially attributed to the function of TIM-3 in mediating T-cell exhaustion. Furthermore, activation of the PI3K/Akt signaling pathway has been identified as a contributing factor [303]. LAG-3 is expressed on activated NK cells, Tregs, B cells, effector T cells (Teffs), and specific DCs. LAG-3 exhibits functional similarities with TIM-3 [182,304,305]. In patients with NSCLC undergoing anti-PD-1 therapy, elevated LAG-3 expression has been linked to diminished therapeutic responsiveness [306]. LAG-3’s interaction with MHC II disrupts the binding between MHC molecules and TCRs on CD4^+^ T cells, directly impeding TCR signaling and subsequent immune responses [307]. Another co-inhibitory receptor, T-cell immunoreceptors with Ig and ITIM domains (TIGIT), is expressed in numerous immune cells [182]. TIGIT overexpression and high infiltration of CD8^+^ TIGIT^+^ T cells have been linked to an unfavorable prognosis in numerous cancer types [308]. Furthermore, mutations within the ITIM domain can disrupt PD-1 signaling and impair T-cell functionality [309]. These findings highlight the complex network of inhibitory mechanisms tumors use to evade immune surveillance. Although current therapies targeting PD-1/PD-L1 interactions show promise, further understanding and targeting these additional co-inhibitory pathways may provide avenues to overcome resistance and improve therapeutic outcomes. The expression of PD-1, CTLA-4, TIM-3, and LAG-3 on the surface of CD8^+^ T cells can be induced by high levels of tumor-derived vascular endothelial growth factor (VEGF), contributing to resistance to anti-PD-1 therapy [310]. The upregulation of VEGFR signaling and thymocyte selection-associated high-mobility group box 1 (TOX) expression exacerbates the activation of inhibitory signaling. TOX is a positive regulator of PD-1, TIM3, TIGIT, and CTLA-4 expression in tumor-infiltrating CD8^+^ T cells. The anti-PD-1 response rates observed in melanoma and NSCLC patients are inversely correlated with the degree of TOX expression in tumor-infiltrating CD8^+^ T cells [310]. In addition, TOX plays a role in the endocytic recycling of PD-1, which is necessary to maintain PD-1, TIM3, TIGIT, and CTLA-4 expression on T-cell surfaces. Moreover, TOX overexpression exacerbates CD8^+^ T-cell exhaustion [311], impeding the efficacy of anti-PD-1 therapy in melanoma and NSCLC patients [312,313].

#### 5.1.5. Deregulated Signaling Pathways in Immune Evasion and ICI Therapy Resistance

The signaling pathways involved in cancer can influence the immune response and contribute to forming an immunosuppressive TME, ultimately facilitating immune evasion [10]. These pathways can alter immune cell composition and cytokine profiles, rendering tumors resistant to ICIs. The loss of tumor suppressor phosphatase and tensin homolog (PTEN), a tumor suppressor, leads to the activation of the phosphatidylinositol 3-kinase (PI3K) pathway, which has been demonstrated to promote tumorigenesis [314] and decrease T-cell function. This is achieved by recruiting inhibitory cells to the TME and expressing VEGF [315,316]. The absence of PTEN has been associated with resistance to anti-PD-1 therapy in uterine leiomyosarcoma [317]. Inhibition of the PI3K-γ pathway reduces myeloid-derived suppressor cells (MDSCs) within the TME, enhancing the efficacy of immune-checkpoint inhibitor (ICI) therapy in animal models [283]. The Wnt/β-catenin signaling pathway plays a role in numerous cellular processes [318] and is frequently dysregulated in tumors, which can cause increased invasiveness and metastatic potential [319]. Constitutive Wnt signaling and stabilization of β-catenin may also contribute to ICI resistance by excluding T cells from the TME. Previous research has indicated a negative correlation between β-catenin levels and TILs, which is attributed to reduced CCL4 expression and impaired recruitment of CD103^+^ DCs, which are crucial for T-cell priming [320]. The mitogen-activated protein kinase (MAPK) pathway involves several cellular processes, including proliferation, apoptosis, and cell migration. Abnormal or misregulated expression of this pathway contributes to cancer [321]. Modifications to the RasMAPK pathway have been demonstrated to impede T-cell recruitment and infiltration in TNBC [322]. Preclinical studies have demonstrated that the efficacy of PD-1/PD-L1 blockade is enhanced when combined with MAPK-targeted therapy [322,323,324].

### 5.2. Acquired Resistance

Acquired resistance, called secondary resistance, represents a complex clinical scenario in which a tumor initially responds to immunotherapy but recurs and progresses. This signifies a transition in the tumor’s susceptibility to the treatment, moving from sensitivity to resistance. The capacity of tumors to evade immune-mediated destruction becomes apparent, challenging the efficacy of the treatment regimen. The host immune system, which plays a pivotal role in mediating antitumor effects, can also contribute to the emergence of therapy resistance. This is due to various adaptive changes in the TME caused by immune pressure. Such alterations include genetic and epigenetic modifications within the tumor, alterations in the tumor, changes in the immune cell landscape, and modifications in cytokine and chemokine profiles. Collectively, these mechanisms diminish immune surveillance and responsiveness, enabling tumor evasion. It is paramount to gain insight into the mechanisms of acquired resistance to develop next-generation immunotherapies and combination treatments that enhance the durability of cancer immunotherapy outcomes (Figure 7).

#### 5.2.1. Interferon-Driven Immune Response and Resistance Mechanisms

Interferons (IFNs) are pivotal as essential cytokines that orchestrate an effective immune response against tumors. IFNs exert their antitumor immune effects by augmenting immune cells’ innate and adaptive functions, primarily through initiating intracellular changes via the Janus kinase (JAK)/signal transducer and activator of transcription (STAT) pathway [325]. Upon recognition of tumor antigens, T-cell-derived IFN-γ initiates signaling via the IFN-γ receptor. This activation results in Janus kinase 1/2 (JAK1/2) activity and the subsequent phosphorylation of signal transducers and activators of transcription (STAT), leading to the upregulation of IFN-γ-stimulated genes [326,327]. Most of these genes are involved in antitumor immunity. However, some genes, such as *CD274* (encoding PD-L1), have the opposite effect of inactivating tumor-specific T cells [66]. This illustrates the dual nature of these genes, which can be beneficial or detrimental. Furthermore, chronic IFN exposure exerts selective pressure on tumors, enhancing immune evasion. During this process, tumors express inhibitory receptors such as PD-L1/2, CTLA-4, or the immunosuppressive metabolite IDO, representing adaptive immune resistance mechanisms. This generally prevents chronic inflammatory processes. However, in the context of cancer, it functions as a mechanism for immune evasion. The downregulation of JAK inhibitors maintains the activation of the JAK/STAT3 pathway, which facilitates tumor growth [328]. However, chronic IFN-γ exposure has been demonstrated to have adverse effects, including the induction of further mutations in tumor cells and upregulation of PD-L1 expression [329]. Tumor-intrinsic mutations that disrupt IFN-γ signaling have been demonstrated to impart resistance to ICIs. A retrospective analysis of patients with melanoma and CRC who did not respond to anti-CTLA-4 or anti-PD-1 therapy revealed these patients had tumors with gene mutations, including JAK1/2 and IFN-γ receptor 1/2 (IFNγR1/2), which are critical for IFN-γ signaling [330,331]. The aforementioned mutations result in a deficiency in PD-L1 expression, even in the presence of IFN-γ, rendering PD-1/PD-L1 blockade therapy ineffective. JAK1/2 also regulates the expression of C-X-C motif chemokines (CXCL), including CXCL9, CXCL10, and CXCL11, which attract T cells. Consequently, tumors with JAK1 loss-of-function mutations were observed to have reduced T-cell infiltration. For example, in patients with melanoma and CRC patients with elevated TMB who did not respond to anti-PD-1 therapy, homozygous loss-of-function mutations in JAK1/2 were identified, which resulted in deficient PD-L1 expression and reduced T-cell infiltration [330].

#### 5.2.2. Immunoediting and Selection Pressure as Mechanisms of Acquired Resistance

The concept of immunoediting posits that the enhanced immunity afforded by anti-PD1/PDL1 therapy safeguards the host against tumor progression and engenders the emergence of tumor subclones that can evade antitumor immunity, precipitating acquired resistance to anti-PD1/PDL1 therapy [194,332,333]. Modifying the DNA copy number in tumor cells can potentially cause antigenicity loss. These cytotoxic T lymphocytes (CTLs) produce interferon-gamma (IFN-γ), which creates an inflammatory environment that can cause melanoma cells to dedifferentiate reversibly. This dedifferentiation causes the loss of melanoma-specific antigens, rendering tumor cells less recognizable to the immune system. This phenomenon indicates that genetic alterations affect neoantigens in response to tumor-specific immune activity. As discussed in a study by Landsberg et al. (2012), inflammation-induced reversible dedifferentiation represents a mechanism by which melanomas evade T-cell therapy, underscoring the dynamic interplay between tumor cells and the immune microenvironment [334,335]. Patients with tumors who have developed acquired resistance to immunotherapy have been observed to lose neoantigens associated with specific mutations [336]. This finding suggests that immunoediting may cause acquired resistance to anti-PD-1/PD-L1 therapy. A matched analysis of pre- and post-treatment tumor tissues from patients who developed resistance revealed the loss of 7–18 putative mutation-associated neoantigens due to genomic modification. These findings support the hypothesis that neoantigen loss is the mechanism underlying ICI resistance [336]. The pathways implicated in primary resistance are also involved in acquired resistance, as they are essential for establishing and maintaining an effective antitumor response. In some patients with melanoma, relapsed tumors develop β2M frameshift deletions during treatment, rendering them resistant to ICIs and invisible to CD8^+^ T cells [337]. Similarly, disruptions in MHC I presentation have been observed in patients with lung cancer, indicating that dysregulated antigen presentation is a critical factor in the development of acquired resistance [250]. In follow-up samples obtained from patients with melanoma who experienced tumor progression after an initial positive response, a homozygous mutation in JAK1/2 was identified. The mutation indicates the selection and proliferation of cancer cells with inactive JAK following the administration of PD-1 blockade therapy. The recurrent tumors exhibited a loss of IFN-γ response, indicating resistance to its cytostatic actions [337]. Recent genetic screenings of tumor cells have illuminated the crucial role of the tumor response to IFN-γ in determining its sensitivity to immunotherapies [338,339]. Within the TME, persistent IFN-γ exerts a dual effect. However, its beneficial effect, often called the “bright side”, may not be sufficient to eliminate tumor cells effectively. Conversely, its negative influence, called the “dark side”, can cause the selection or emergence of tumor clones exhibiting a more aggressive phenotype. This can ultimately result in the development of resistance to immunotherapeutic interventions [340]. This observation is consistent with previous research indicating that sustained IFN-γ signaling in tumor cells can activate STAT-1-dependent mechanisms, leading to epigenetic and transcriptional modifications within these cells. Due to these modifications, tumor cells express multiple ligands corresponding to T-cell inhibitory receptors. This expression enhances resistance to immune checkpoint therapies, contributing to the challenging landscape of immunotherapy resistance in cancer [329]. Moreover, a previous study demonstrated that individuals with metastatic melanoma treated with a combination of anti-CTLA-4 and anti-PD-1 therapies exhibited activation of the β-catenin signaling pathway and deletion of the PTEN gene, two oncogenic abnormalities associated with insufficient T-cell infiltration into tumor sites. This ultimately contributed to the development of the acquired resistance [341].

#### 5.2.3. Alternative Immune Checkpoint Activation and Acquired Resistance Mechanisms

Anti-PD-1/L1 and anti-CTLA-4 therapies target a specific subset of immune checkpoints (ICs), allowing tumors to evade immune surveillance by activating alternative inhibitory signals. Upregulation of LAG-3, TIM-3, TIGIT, and VISTA has been observed in patients with tumor recurrence after anti-PD-1/L1 or anti-CTLA-4 therapy, indicating the presence of an underlying mechanism of acquired resistance [342]. A subset of patients with melanoma who express MHC II have been reported to respond favorably to anti-PD-1 therapy [343]. However, LAG-3 upregulation in tumor-infiltrating lymphocytes (TILs) in MHC II^+^ tumors has been shown to result in the failure of anti-PD-1 treatment following an initial response [343,344]. Upregulation of TIM-3 has been observed in TILs in preclinical and clinical studies following treatment, leading to acquired resistance to the PD-1/PD-L1 blockade [183,185,345]. Similarly, the expression of TIM-3 is increased following treatment with anti-PD-1 antibodies in melanoma and NSCLC, which may contribute to the emergence of acquired resistance to PD-1/PD-L1 blockade [344].

#### 5.2.4. Impaired CD8^+^ T-Cell Memory and Acquired Resistance

The role of T-cell memory, particularly that of tissue-resident memory (TRM) CD8^+^ T cells, is pivotal in mediating antitumor immunity and influencing resistance to ICIs [346]. To optimize the efficacy of immunotherapies, it is imperative to elucidate the specific biological processes and environmental conditions that facilitate the generation and sustained maintenance of CD8^+^ T-cell immunological memory that targets cancer cells. This includes investigating how these memory cells recognize and respond to tumor antigens, the signaling pathways supporting their long-term survival and functional capacity in the TME, and how they can be effectively bolstered or reactivated in immunotherapy. A comprehensive understanding of these aspects is essential, as robust and durable CD8^+^ T-cell memory can lead to long-lasting, complete responses to cancer treatment, transforming the landscape of patient outcomes. The pivotal role of TRM CD8^+^ T cells is supported by research involving patients with BC, in which TRM CD8^+^ T cells have been demonstrated to facilitate immune surveillance [347].

Similarly, patients with melanoma who exhibited a positive response to anti-PD-1 therapy demonstrated an increased population of CD8^+^ effector memory T cells [324]. Epigenetic and genomic investigations have revealed the presence of a subset of progenitor-exhausted memory cells in CD8^+^ TILs, which exhibit characteristics reminiscent of stem cell-like memory cells. These cells express TCF-1 and have been associated with favorable clinical outcomes following anti-PD-1 therapy [348]. Furthermore, whereas CD8^+^ T cells are essential for immune-mediated tumor rejection, given their correlation with patient survival and treatment success, it is crucial to recognize that the collaboration of various immune cells with CD8^+^ T cells is likely essential for a successful and enduring antitumor immune response [349,350]. Therefore, a comprehensive understanding of T-cell memory generation and persistence is imperative to improve patient selection for ICI therapy. T-cell memory generation involves the development of long-lived memory T cells, activated upon exposure to an antigen, that respond rapidly to subsequent encounters with the same antigen. The initial strength of T-cell receptor (TCR) signaling, the presence of co-stimulatory signals, the cytokine environment, and the metabolic status of T cells are crucial factors that influence T-cell memory [346]. The capacity of T cells to persist within the host for an extended duration is defined as T-cell persistence. This process is influenced by many factors, including the expression of survival markers (such as Bcl-2), telomere length maintenance, and chronic activation prevention, which can induce T-cell fatigue. A more comprehensive understanding of these mechanisms will facilitate the determination of the type, intensity, and duration of tumor responses to ICIs. Additionally, identifying patients exhibiting deficiencies in T-cell memory and persistence will facilitate the prediction of patients likely to develop resistance to ICI therapy [346].

## 6. Therapeutic Strategies for Overcoming Drug Resistance with ICIs

Given the various factors contributing to ICI resistance, many strategies that have been devised to counteract this resistance by targeting the underlying mechanisms focus on various stages of the cancer-immunity cycle to prevent tumors from evading immune surveillance. Combination therapies are often necessary to achieve enhanced clinical outcomes [351]. These strategies encompass increasing T-cell infiltration, overcoming T-cell exhaustion, BITEs, epigenetic modulation, improving the TME, and enhancing T-cell priming. A summary of these strategies is provided in Table 2, followed by a discussion of each strategy.

### 6.1. Enhancing Tumor Infiltration by T Cells

Adequate T-cell infiltration must occur to reinvigorate antitumor immunity via PD-1/PD-L1 blockade. Combined with ICIs, ACT therapy has been demonstrated to enhance TILs and T-cell cytotoxicity and surmount the constraints imposed by MHC dysfunction [365,366], as evidenced by its favorable clinical outcomes in hematologic malignancies [366]. Nevertheless, this approach’s efficacy in treating solid tumors remains unclear. In several murine models, various ACT approaches, with PD-1/PD-L1 and CTLA-4 blockade, have been observed to induce tumor regression [187,365]. A preclinical investigation revealed that activation of the lymphotoxin β-receptor resulted in a notable increase in chemokine synthesis, significantly enhancing T-cell recruitment. Furthermore, combined therapy with PD-L1 inhibition has been shown to result in substantial tumor regression [352].

### 6.2. Bispecific T-Cell Engagers as Immunotherapeutics

Bispecific T-cell engagers (BiTEs) constitute a class of engineered bispecific monoclonal antibodies (mAbs) that attract considerable attention as potential anticancer immunotherapeutic agents. These innovative agents are designed with a strategic approach to harness the cytotoxic potential of T cells within the host immune system and redirect their activity toward cancer cells, making them a highly encouraging approach in cancer immunotherapy. The fundamental concept underlying BiTE therapy is the development of bispecific antibody constructs capable of simultaneously binding to a surface molecule expressed on T cells and a specific antigen present on cancer cells [367]. This dual-binding mechanism initiates a cascade of events that result in the lysis of tumor cells, effectively and precisely targeting and eliminating cancer cells. The remarkable efficacy of BiTE therapy, particularly in treating B-cell malignancies, has marked a significant breakthrough in cancer immunotherapy [367].

Nevertheless, several challenges have emerged, including issues like antigen loss and the upregulation of ICMs, which are associated with resistance to BiTE therapy [367]. In response, researchers have proactively adapted antibody constructs and investigated combination strategies to enhance the efficacy of therapy and reduce the associated toxicity [367]. Ongoing efforts are essential for overcoming these limitations and advancing BiTE therapy as a viable and effective treatment option. It is noteworthy that blinatumomab, a CD3/CD19 bispecific antibody, has been granted FDA approval as the sole bispecific T-cell engager. This treatment is specifically indicated for patients with relapsed or refractory B-cell ALL (B-ALL) [368], which underscores the clinical potential of this therapeutic approach.

The potential of BiTE therapy in combating drug resistance has been actively investigated, particularly when combined with ICIs. Recent studies have demonstrated that combining CD30/CD16A biAbs with PD-1 inhibitors enhances clinical and pharmacodynamic activity [369]. Furthermore, this combination therapy has been demonstrated to positively affect immune cell recruitment during the treatment of Hodgkin’s lymphoma [369]. BiTEs have emerged as vital elements in therapeutic strategies with ICIs, enabling the overcoming of drug resistance. They represent a promising approach for redirecting T cells to combat cancer and have demonstrated remarkable efficacy, particularly in treating B-cell malignancies.

### 6.3. Epigenetic Modulation to Reshape the Tumor Microenvironment

Epigenetic regulation has emerged as a promising approach for cancer immunotherapy [370,371]. In the intricate TME, malignant cells use epigenetic processes to circumvent immune-mediated cell death, evade host immune recognition, and avoid immunogenicity. Concurrently, immune cells undergo various epigenetic modifications that affect cytokine expression, cancer immunosurveillance, ICAM expression of ICMs, and presentation of TAAs via MHC molecules. Consequently, targeting epigenetic modifiers to reshape the immune microenvironment represents a significant potential avenue for developing anticancer strategies [372]. Epigenetic alterations have been demonstrated to disrupt AP mechanisms in tumor cells. Studies have indicated that DNA methyltransferases (DNMT) and histone deacetylases (HDAC) can inhibit AP. Targeting of tumor cells with DNA methyltransferase inhibitors (DNMTi) and histone deacetylase inhibitors (HDACi) has been demonstrated to result in the re-expression of MHC class I molecules, which are essential for AP [373,374]. Furthermore, epigenetic mechanisms contribute to the upregulation of PD-L1 expression in tumor cells, which can inhibit the immune response. The family of proteins known as the Bromodomain and Extra-Terminal (BET) proteins plays a crucial role in regulating epigenetic gene expression. These proteins can identify and attach acetylated lysine sites on histones within nucleosomes, facilitating the recruitment of transcriptional machinery to regulate gene expression [375]. Inhibition of BET proteins may effectively limit tumor progression and enhance antitumor immune responses, as it can prevent IFN-γ-induced PD-L1 expression [376,377]. Circulating natural killer (NK) cells play a role in determining the spread of cancer cell metastasis. The activating receptor NKG2D can recognize a variety of ligands present in cancer cells, which is crucial for the ability of NK cells to lyse tumors. EZH2, a histone methyltransferase, has been demonstrated to promote lineage commitment and functional differentiation of NK cells while simultaneously suppressing the expression of activating NK receptors, such as NKG2D [372]. Distinct HDAC inhibitors (HDACis) may elicit disparate effects on NK cell functional phenotypes. Non-selective HDACis, including valproic acid (VA), trichostatin A (TSA), and sodium butyrate, have been demonstrated to impair NK cell-mediated cytotoxicity and downregulate the activating receptor NKG2D [378,379]. In contrast, specific class I HDAC inhibitors such as entinostat have been demonstrated to promote NKG2D expression and NK cell activation [380]. Epigenetic variations within T-cell populations can cause disparate levels of T-cell activation and sensitivity to related signaling pathways. Another strategy for enhancing the effectiveness of ICIs is to manipulate these epigenetic states to enable functional T-cell activation, and heightened sensitivity is another strategy to enhance the effectiveness of ICIs [381]. Furthermore, epigenetic mechanisms have been shown to contribute to cell exhaustion [382,383]. The histone deacetylase (HDAC) inhibitor valproic acid (VPA) has demonstrated the capacity to reverse the functional state of exhausted T cells. Epigenetic drugs have been investigated for their potential to enhance the efficacy of cancer immunotherapy in several ways. Recent studies have focused on combining IC blockade therapy with epigenetic modulation, representing a noteworthy advancement in this field. The Food and Drug Administration has approved the use of classical epigenetic drugs, including histone deacetylase inhibitors (HDACis) and DNA methyltransferase inhibitors (DNMTis), for cancer treatment [384]. The concomitant administration of DNA methyltransferase (DNMT) and histone deacetylase inhibitors (HDACi) augments the efficacy of anti-programmed death-1 (anti-PD-1) and anti-cytotoxic T-lymphocyte-associated protein 4 (anti-CTLA-4) therapies in colorectal cancer (CRC) and breast cancer (BC) models by reducing myeloid-derived suppressor cells (MDSCs) [354]. Several clinical trials are in progress to evaluate the safety and efficacy of combining epigenetic modulation with ICI strategies [187,385].

### 6.4. Improving the Tumor Immune Microenvironment

The isTME has been demonstrated to impede the functionality of CD8^+^ T cells, engendering resistance to anti-PD-1/PD-L1 therapy. Targeting TAMs enriched in the TME has been demonstrated to improve the response to treatment [386]. A preclinical study reported that the simultaneous blockade of colony-stimulating factor 1 receptor (CSF1R) and anti-PD-1 antibodies resulted in reprogramming M2 TAMs into M1 and promoting tumor regression [273]. Metabolites such as indoleamine 2,3-dioxygenase (IDO) and adenosine play pivotal roles in modulating tumor immunity. The inhibition of these metabolites has been demonstrated to enhance antitumor immune responses when combined with anti-programmed death-1 (PD-1, PD-L1) and anti-cytotoxic T-lymphocyte-associated protein 4 (CTLA-4) therapies [355,387]. Research has demonstrated the effectiveness of combining CD73 inhibitors to halt adenosine production or A2A receptor blockade to interfere with its signaling with anti-PD-1/PD-L1 treatment, resulting in antitumor efficacy [356,388]. In addition, immunosuppressive cytokines derived from tumors have been shown to promote resistance to ICIs [288]. Nevertheless, inhibition of TGF-β can potentially reverse the isTME, facilitate T-cell infiltration, and enhance the efficacy of anti-PD-1 therapy [357]. Inhibition of CXCR2 [282] and CXCR4 [292] chemokine receptors has been demonstrated to disrupt the trafficking of MDSC and Treg cells, increasing their sensitivity to PD-1/PD-L1 blockade. Moreover, vascular endothelial growth factor (VEGF) inhibitors have demonstrated efficacy in restoring the immune-suppressed TME and reversing immunotherapy resistance [389]. The Food and Drug Administration (FDA) approved bevacizumab combined with atezolizumab and chemotherapy after demonstrating enhanced progression-free survival (PFS) and overall survival (OS) in patients diagnosed with metastatic non-small cell lung cancer (NSCLC) [115]. In addition, PI3K activation has been linked to the formation of an isTME. Consequently, the inhibition of PI3K has been proposed as a potential strategy to enhance the efficacy of ICI therapy. Preclinical studies have demonstrated encouraging outcomes with the triple blockade of CTLA-4, PD-1, and PI3K [358,390].

The combination of ICIs with targeting of tumor metabolism represents an emerging approach in cancer research with the potential to enhance cancer treatment outcomes [391]. This strategy is based on the understanding that tumor cells undergo metabolic reprogramming, establishing altered metabolic pathways supporting their rapid growth and survival [392]. By targeting these metabolic vulnerabilities combined with ICIs, researchers have aimed to enhance the effectiveness of immunotherapy and overcome resistance to treatment. Enhanced glucose metabolism in tumor cells has been demonstrated to promote unchecked cellular proliferation and control the expression of inhibitory ICMs, including PD-L1 [393]. Furthermore, metabolic alterations (glucose metabolism, lipid metabolism, and amino acid metabolism) in T cells within the TME have been emphasized to elucidate the interplay between IC signaling and metabolism in T cells [394].

### 6.5. Increasing Tumor Immunogenicity and T-Cell Priming

Tumor immunogenicity is defined as the capacity of a tumor to provoke an immune response [395]. Highly immunogenic tumors can elicit a robust immune response. In contrast, tumors with low immunogenicity may evade detection and elimination by the immune system [396]. Various strategies have been developed to enhance tumor immunogenicity and T-cell priming, including the following.

#### 6.5.1. Oncolytic Virus Therapies and Vaccines Against Cancers

Combining cancer vaccines and immune-checkpoint inhibitor (ICI) therapy represents a promising strategy for enhancing antitumor immunity. The use of tumor-specific peptides or DCs in cancer vaccines, combined with immune-checkpoint inhibitor (ICI) therapy, has been demonstrated to prime T cells. Early studies in melanoma patients have demonstrated that the combination of a dual multi-peptide vaccine and nivolumab is associated with improved survival outcomes [359]. Recently, patients with melanoma who received personalized tumor neoantigen vaccination and anti-PD-1 therapy exhibited a complete response [397], which further supports the further development of vaccines against cancers and oncolytic virus therapies. Oncolytic virus therapy (OVT) attracts T cells to the tumor site, alters the tumor microenvironment (TME), and inhibits tumor angiogenesis, providing another potential therapeutic strategy. Combining talimogene laherparepvec (T-VEC) with ipilimumab [146] or pembrolizumab [147] has been more efficacious than monotherapy. Numerous clinical trials combining ICI with T-VECs or other novel oncolytic viruses are underway.

#### 6.5.2. Chemotherapy

It is conceivable that CT may induce immunogenic cell death (ICD) in tumor cells, resulting in the release of tumor antigens and initiation of damage-associated signals. This process enhances CTL priming, rendering tumors more responsive to ICIs [398]. This process transforms DCs into immunostimulatory antigen-presenting cells (APCs), facilitating CD8^+^ T cells [399]. Specific chemotherapeutic agents such as decitabine have been observed to enhance the expression of tumor-associated antigens, including melanoma-associated antigen 3 (MAGE-A3), rendering ESCC cells more visible to the immune system. Chemotherapeutic agents such as gemcitabine and cyclophosphamide have been demonstrated to deplete myeloid-derived suppressor cells (MDSCs) and regulatory T cells (Tregs) while also effectively targeting and killing tumor cells [400,401]. Prior research has demonstrated the efficacy of combining chemotherapy with PD-1/PD-L1 blockade in patients [191,360,398,399].

#### 6.5.3. Radiotherapy

RT has long been acknowledged as a highly efficacious modulator of the TME. The induction of DNA damage destroys tumor cells, subsequent release of tumor antigens, and production of inflammatory mediators. This heightened state of inflammation within the TME enhances the immunogenicity of the tumor, creating an opportunity to combine RT with other therapeutic agents, such as ICIs, to capitalize on this immune activation [402]. Furthermore, RT has been demonstrated to enhance the activation of DCs and augment the production of proinflammatory cytokines, stimulating an increase in TILs [403]. Thus, the function of TLRs is important. TLRs, including TLR3 and TLR9, are components of the innate immune system that serve as sensors and recognize damage-associated molecular patterns (DAMPs) and pathogen-associated molecular patterns (PAMPs). DNA damage caused by RT in tumor cells results in the release of damage-associated molecular patterns (DAMPs). These DAMPs are recognized by TLRs, which play a pivotal role in the maturation and activation of DCs [362,404]. This process enhances the functionality of DCs, which is crucial for overcoming resistance to anti-PD-1/PD-L1 therapy.

Radiotherapy (RT) provides a sophisticated interplay between immune responses within the TME. Following irradiation, tumor cells release damage-associated molecular patterns (DAMPs) and cytokines, which stimulate the production of type I interferons, particularly IFN-α [131]. Consequently, RT serves as an initial stimulant of the TME, enhancing tumor immunogenicity and triggering a cascade of immune responses. This encompasses the modulation of TLRs and the production of interferon-alpha (IFN-α), which can be augmented by introducing an IFN-α fusion protein. These combined actions serve as successive layers of immune enhancement, each of which contributes to the optimization of the therapeutic efficacy of ICIs. Consequently, radiotherapy serves not only as a direct treatment for tumors but also as a catalyst for immune-mediated antitumor effects [405].

### 6.6. Combination with Other Targeted Therapies

Several targeted therapies are being investigated for use with ICIs. These therapies target canonical oncological signaling pathways that can influence the TME by reducing the production of inhibitory cytokines, enhancing the efficacy of immune-checkpoint inhibitor (ICI) therapy. For example, using BRAF inhibitors to target the MAPK pathway results in increased infiltration of CD8^+^ T cells and increased production of IFN-γ [363]. Early trials of ipilimumab and BRAF/MEK inhibitors were terminated due to an increased incidence of toxicity. Nevertheless, additional preclinical and clinical investigations combining these inhibitors with anti-PD-1/PD-L1 therapy have demonstrated enhanced antitumor immunity and tolerability [187,323,406]. Moreover, patients with BRAF V600-mutated metastatic melanoma who received triple combination therapy targeting BRAF, ERK, and PD-1 exhibited a durable antitumor response [364].

## 7. Comprehensive Biomarkers in Immune-Checkpoint Inhibitor Therapies

ICIs have transformed the field of oncology. Nevertheless, it is essential to note that the long-term benefits of ICIs are observed only in a subset of patients with cancer. This finding highlights the need for further research to fully understand the therapeutic potential of ICIs in a broader range of patients [407]. Moreover, the utilization of ICIs is associated with a considerable disadvantage: the potential for severe autoimmune adverse effects, some of which may be life-threatening. Therefore, the development of reliable and robust biomarkers is imperative. Identifying biomarkers capable of predicting both response and resistance to ICIs is paramount. These biomarkers serve two purposes: first, to identify patients most likely to experience the therapeutic benefits of ICIs, thus ensuring that they receive optimal treatment, and second, to identify those who are less likely to respond. In the latter group, this knowledge enables clinicians to avoid the unnecessary risks and potential toxicities associated with ICI administration [408]. Over time, numerous biomarkers have been proposed, each of which has the potential to guide ICI therapy to enhance the response rates and overall efficacy in patients undergoing immunotherapy with ICIs. These biomarkers encompass a range of factors, including PD-L1 expression, TMB, MSI, TILs, cytokines, transcription factors, and microbiome composition. The aforementioned biomarkers collectively provide invaluable insights into tailoring ICI treatment strategies for individual patients, thus ushering in a new era of personalized and effective cancer immunotherapy [409].

### 7.1. PD-L1 Expression

PD-1 and PD-L1 are extensively researched biomarkers in immunotherapy using ICIs [246,410]. PD-L1 expression is widely recognized as the most commonly influenced immune-based biomarker in clinical practice [411]. Conversely, TLRs represent a subtype of non-catalytic receptor markedly expressed in antigen-presenting cells (APCs) and activated by pathogen-associated molecular patterns (PAMPs). These proteins play pivotal roles in the regulation of PD-L1 [412]. Specifically, the TLR-mediated regulation of PD-L1 depends on the activation of MEK/ERK kinases, which enhance PD-L1 messenger RNA (mRNA) transcription via nuclear factor kappa B. In addition, IFN-γ receptors 1 and 2 have been identified as key regulators of PD-L1 expression, primarily through Jak/STAT-mediated activation of IRF-1. In addition, IFN-mediated activation of Jak/STAT can upregulate PD-L1 expression through the MEK/ERK and phosphatidylinositol 3 kinase (PI3K)/AKT pathways, which facilitates PD-L1 transcription through phosphorylation of the mammalian target of rapamycin [413].

During carcinogenesis, the expression of PD-L1 can be increased because of oncogenic drivers. For example, there is a positive correlation between EGFR mutations and PD-L1 expression in lung cancer, and EGFR inhibitors act as repressors of PD-L1 transcription [414]. In tumors with phosphatase and tensin homolog (PTEN) mutations, PD-L1 overexpression is maintained by uncontrolled activation of the PI3K/AKT pathway [415]. Moreover, in T-cell lymphoma, the nucleophosmin (NPM)/anaplastic lymphoma kinase (ALK) fusion gene has been demonstrated to upregulate PD-L1 expression via constitutive STAT3 activation [416]. These pathways underscore the significance of PD-L1 as a biomarker for predicting responses to ICIs. Patients that overexpress PD-L1 will likely demonstrate a more favorable prognosis and benefit from ICIs [417,418,419]. Several studies have demonstrated that patients with tumors exhibiting positive PD-L1 expression scores show higher response rates and greater therapeutic benefits to ICIs across various cancer types, including NSCLC, UC, and others [9,420,421]. For instance, patients with non-small cell lung cancer (NSCLC) who exhibit high levels of PD-L1 expression usually derive the most significant benefit from pembrolizumab, as evidenced by phase I study 430 findings. In the CheckMate 012 study, patients with advanced NSCLC were treated with a combination of nivolumab and ipilimumab, and excellent response rates were documented in patients with PD-L1-positive tumors [422,423]. Consequently, PD-L1 expression remains an imperfect indicator for predicting the efficacy of anti-PD-1/PD-L1 therapy [424]. Some clinical studies have demonstrated an inadequate direct correlation between PD-L1 expression and treatment effect [425]. Anti-PD-1/PD-L1 therapy has been demonstrated to confer benefits to patients with low PD-L1 expression, indicating that its effectiveness is not limited to individuals with high PD-L1 levels [115,426,427,428]. In patients with NSCLC, studies such as CheckMate 017 and CheckMate 057 indicated a positive correlation between PD-L1 expression of 50% and a more significant overall survival (OS) benefit from nivolumab. Nevertheless, individuals with as little as 1% PD-L1 expression demonstrated effective responses to treatment [429].

### 7.2. Tumor Mutational Burden

Tumor mutational burden (TMB) was defined as the total number of non-synonymous mutations in a tumor per megabase [430]. It has been demonstrated that TMB can potentially trigger T-cell activation via neoantigen production [431,432]. TMB can serve as a biomarker for predicting the responsiveness to ICB therapy [245]. Tumors with high TMB levels have demonstrated improved clinical efficacy of PD-1/PD-L1 Ab treatment in a range of cancer types, including NSCLC [245,433], UC [420], and others [242,324,434,435]. High TMB is defined as TMB ≥ 10 mut/Mb and has been linked to improved survival in several tumor types [436]. A retrospective analysis of 27 cancer types revealed a positive correlation between TMB and the therapeutic efficacy of PD-1/PD-L1 antibodies and an increased ORR [437]. The combination of nivolumab and ipilimumab yielded superior overall responses (ORs) in NSCLC patients with TMB levels equal to or greater than 10 mut/Mb, despite PD-L1 expression [438]. However, tumors with low TMB, such as clear cell RCC (ccRCC) and virus-driven tumors, have been observed to exhibit significant response rates to ICI therapy [437,439], indicating that TMB may not be the sole determinant of response. Detecting TMB using next-generation sequencing (NGS) or polymerase chain reaction (PCR) is a relatively expensive and time-consuming process that limits its clinical applicability as a biomarker.

### 7.3. Microsatellite Instability and Mismatch Repair Deficiency

Mismatch repair (MMR) pathways are vital for identifying and repairing mismatched bases during DNA replication and gene recombination [94]. Deficiencies in this system, such as MMR-D, can cause hypermutations in cancer cells and increased susceptibility to mutations in repetitive DNA sequences, which may lead to MSI-H in some cancers [440,441]. MSI status can potentially serve as a biomarker for cancer immunotherapy, particularly in CRC, GC, and endometrial cancers (EndC) [442]. It has been proposed that MMR-D may serve as a predictive biomarker in immunotherapy using ICIs, enabling the assessment of a tumor’s response to treatment [197]. A higher response rate to ICIs has been observed in colon cancers with mismatch repair deficiency (MMR-D) than in those with intact MMR [197].

Additional research has demonstrated that tumors with MMR-D and elevated MSI-H levels exhibit enhanced responsiveness to pembrolizumab compared to those with intact MMR-proficient tumors [101,443]. In 2017, the Food and Drug Administration (FDA) approved the use of pembrolizumab to treat advanced solid tumors with MSI-H or DNA MMR-D [444]. MSI-H tumors have a higher neoantigen load, which can activate lymphocytes and increase their sensitivity to ICIs [444]. Two principal methodologies were used to identify MMR-D in the tumor specimens. Immunohistochemistry (IHC) was used to survey specific MMR protein expression, whereas direct assays for MSI were conducted through DNA sequencing, using either PCR or NGS [445]. Furthermore, MMR-D is associated with high TMB and heightened sensitivity to ICI therapy [446]. Patients with MMR-D or MSI-H metastatic colorectal cancer (mCRC) who possess lower TMB values may experience early disease progression when treated with ICIs. Conversely, patients with the highest TMB values may benefit most from intensified combination therapy involving anti-CTLA-4 and PD-1 inhibitors [447].

### 7.4. Cytokines and Transcription Factors

Although the utilization of IFN-γ as a biomarker to augment the response to immunotherapy in patients undergoing ICI therapy remains a topic of active investigation, several studies have offered insights into its potential utility. For example, one study reported that melanoma patients with high pretreatment levels of IFN-γ in the TME exhibited a significantly greater objective response to anti-PD-1 therapy [448]. Similarly, it was reported that NSCLC patients with non-small cell lung cancer (NSCLC) who exhibit elevated baseline levels of interferon-gamma (IFN-γ) in their peripheral blood exhibit prolonged progression-free survival (PFS) upon treatment with PD-1 inhibitors [449]. Furthermore, monitoring IFN-γ levels during treatment can provide insights into the treatment response. For instance, research has shown that increased IFN-γ production during therapy is associated with favorable outcomes in melanoma patients receiving ICIs [450]. Furthermore, IFN-γ levels can inform the selection of combination therapies to enhance response [137]. In conclusion, IFN-γ is a promising biomarker for optimizing immunotherapy responses. However, further research must validate its clinical utility and establish standardized guidelines for its use [451].

CD163 expression is exclusive to the monocyte–macrophage lineage, with elevated levels of expression observed in red pulp macrophages, bone marrow macrophages, liver macrophages (known as Kupffer cells), lung macrophages, and macrophages in various other organs. It is involved in the clearance of hemoglobin–haptoglobin complexes and plays a role in the resolution of inflammation. CD163 has been identified as a potential biomarker for ICI therapy in patients with cancer. It is associated with improved overall survival (OS) [452] and progression-free survival (PFS) in patients receiving ICIs [453]. Further research must ascertain the clinical utility of CD163 as a biomarker of ICI therapy [452]. The optimal timing for assessing CD163 as a predictive marker in cancer management is contingent on the specific clinical context and evaluation objectives. For predictive purposes, the optimal timing for assessing CD163 levels in tumor tissue is before the commencement of cancer treatment. This preliminary assessment has the potential to provide crucial insights into patient outcomes and probable responses to therapy. Specifically, elevated levels of CD163^+^ TAMs before treatment initiation may indicate an increased likelihood of treatment resistance [454]. For example, in a specific study, serum concentrations of soluble CD163 (sCD163) were meticulously quantified at baseline (day 0) and 42 days after nivolumab treatment. A statistically significant variation in serum levels was observed on day 42 compared to baseline. This observation underscores the potential value of sCD163 as a dynamic marker, particularly in the context of specific immunotherapies [455]. Other cytokines, including granulocyte colony-stimulating factor (G-CSF), granulocyte-macrophage colony-stimulating factor (GM-CSF), fibroblast growth factor (FGF)-2, IFN- α 2, IL-2, and IL-13, which are upregulated before or shortly after ICI treatment, have also been demonstrated to predict irAEs [233]. The presence of immune cells and cytokines involved in the immune-killing process can be used as an indicator of the success of immunotherapy and irAEs.

### 7.5. Tumor-Infiltrating Lymphocytes

TILs have been demonstrated to be valuable tools for predicting favorable prognosis. The subject of ongoing research is immune infiltration by TILs in terms of phenotype, distribution, and complexity. The CD8^+^ TIL score is particularly relevant because higher scores are associated with more pronounced clinical benefits of immunotherapy [450,456]. Furthermore, the presence of CD8^+^ TILs is correlated with PD-L1 expression, and this correlation is considered a predictor of response to ICIs. Initial research on metastatic melanoma demonstrated that CD8^+^ T cells at the tumor margins were associated with a favorable response to pembrolizumab [448]. However, CD8^+^ TILs exhibit significant heterogeneity, with only a small fraction capable of recognizing tumor mutation-related antigens. The remainder are classified as “bystander T cells”, insensitive to tumors. The aforementioned distinct populations were differentiated based on CD39 expression levels [457]. A more precise predictor of a patient’s response to ICIs may entail consideration of both CD39 expression and the number of CD8^+^ TIL cells [458].

### 7.6. Microbiome and Immunotherapy

The microbiome influences the response to ICI therapy, mainly owing to its profound impact on the immune system [459]. The microbiome can enhance immune responses, trigger inflammation, and disrupt the delicate equilibrium between cell proliferation and cell death, potentially promoting tumorigenesis [460]. The gut microbiome has demonstrated that it can activate T-cell-mediated responses by directly targeting tumor cells [461,462]. In vivo experiments with mice bearing CT26 tumors have demonstrated that a diverse microbial population within the gut markedly enhances the response to ICI therapy by increasing the secretion of critical cytokines, such as IL-2 and IFN-γ, compared to mice treated with antibiotics [460]. Furthermore, research has elucidated a definitive correlation between the composition of the human gut microbiome and the response to ICIs in patients with cancer. For example, melanoma patients receiving anti-PD-1 therapy exhibit elevated levels of specific microbial species, including *Bifidobacterium longum*, *Collinsella aerofaciens*, and *Enterococcus faecium*, highlighting the microbiome’s relevance in ICI response [463]. Disruption of the gut microbiome through antibiotics has been linked to a diminished response to ICIs in mouse models [464]. 

Clinical studies have documented a correlation between prior antibiotic use and diminished responsiveness to ICI therapy [465,466]. Despite the long-held perception that healthy lungs are devoid of microbes, recent research has uncovered diverse microbial communities within the lungs. These include members of *Bacteroidetes*, *Proteobacteria*, *Firmicutes*, and *Actinobacteria*, identified as commensal microbiota [467,468]. These commensal microbiota influence immune tolerance and mitigate lung inflammation through interactions with dendritic cells, gamma delta T cells, and regulatory T cells [469]. Several clinical studies have demonstrated a correlation between lung microbiota and cancer. A dominant hypothesis posits that lung microbiota may directly contribute to carcinogenesis by intensifying mucosal inflammation and fostering immunological dysfunction [470]. Dysbiosis in the lungs is defined as an aberrant alteration in the migration, elimination, and growth of microbial features, ultimately affecting lung microbiota composition and function [471]. This can disrupt the baseline activity of the immune system [472,473]. Dysbiosis can cause excessive immune system activation, uncontrolled expansion of IL-17-producing CD4 helper T cells, and compromised APC priming. This impairment impairs the ability of APCs to respond to tumor antigens, which is a crucial step in the development of lung tumors [474,475].

It has been demonstrated that the commensal microbiota can considerably influence the efficacy of immunotherapeutic interventions in human cancers [466,476]. While the relationship between the respiratory microbiota and the response to ICIs in lung cancer remains relatively understudied, it is noteworthy that the respiratory microbiota has been observed to induce inflammation linked to lung cancer [477].

## 8. Conclusions and Future Perspectives

Significant advancements in IC modulation have profoundly transformed the oncological landscape, providing novel and targeted strategies for cancer treatment. The deployment of ICIs has demonstrated notable efficacy by targeting pivotal immune regulatory checkpoints, including PD-1, PD-L1/L2, CTLA-4, and LAG-3. This approach exploits the potential of the immune system to recognize and eradicate malignant cells. Notwithstanding the considerable headway made in cancer immunotherapy, considerable hurdles remain to be overcome to surmount resistance mechanisms and identify predictive biomarkers for personalized therapy. Although some patients experience a notable reduction in tumor size following ICI therapies, many patients and tumors do not respond to these treatments, a phenomenon known as primary resistance. Moreover, some patients initially respond to immunotherapy but eventually develop resistance and experience tumor relapse, a phenomenon known as acquired resistance. A comprehensive understanding of these resistance mechanisms is essential to close the current knowledge gap and advance the field of cancer immunotherapy. The development of reliable and robust biomarkers is a fundamental objective for the optimization of ICI therapy. Such biomarkers can facilitate the selection of patients most likely to benefit from treatment and enable the implementation of personalized therapeutic approaches. New biomarker development approaches include assessing PD-L1 tumor expression, TIL status, and TMB. Nevertheless, existing data on biomarkers for irAEs remain limited and controversial, thus necessitating further investigation. Recent advancements in cancer immunotherapy have fostered interest in exploring novel ICI targets and combination therapies to enhance treatment efficacy and overcome drug resistance. The combination of immunotherapy with targeted therapies, chemotherapy, and radiation therapy has the potential to yield promising synergistic effects that could ultimately enhance patient outcomes.

Artificial intelligence (AI) and machine learning are rapidly emerging as indispensable tools in the field of oncology with the potential to revolutionize the discovery and implementation of predictive biomarkers. The application of artificial intelligence (AI) in the analysis of complex multi-omics data has the potential to provide invaluable insights into tumor biology, immune interactions, and resistance mechanisms. The application of AI enables researchers to expedite the discovery of novel biomarkers, forecast treatment outcomes, and refine personalized cancer immunotherapy. Integrating AI, advanced imaging techniques, and single-cell sequencing technologies will facilitate real-time monitoring of the dynamic interplay between tumors and the immune system, elucidating the intricate mechanisms that drive resistance and response. By elucidating intricate tumor–immune interactions and identifying novel targets, scientists can devise innovative therapeutic strategies to circumvent resistance and enhance the efficacy of cancer immunotherapy.

In conclusion, the rapidly evolving field of IC modulation has transformed oncology, offering unprecedented optimism in the fight against cancer. Notwithstanding the persistent challenges in comprehending the resistance mechanisms and identifying predictive biomarkers, the unwavering commitment to research and innovation bodes well for the future of personalized cancer immunotherapy. The combination of artificial intelligence-driven techniques, novel ICIs, and synergistic combination therapies will propel the field toward new frontiers, ultimately transforming the lives of cancer patients worldwide.

## Figures and Tables

**Figure 1 cancers-17-00880-f001:**
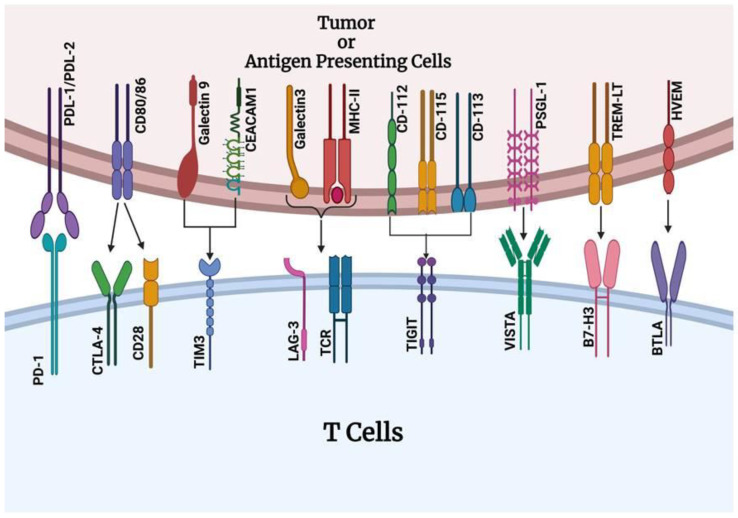
Signal Transduction Pathways of Co-inhibitory Immune Checkpoints. This figure illustrates the complex signaling pathways mediated by various co-inhibitory immune checkpoints on T cells and their interactions with the corresponding ligands on APCs and tumor cells. The immune checkpoints depicted include the following: PD-1 interacts with PD-L1 and PD-L2, CTLA-4 binds to CD80 (B7-1) and CD86 (B7-2), LAG-3 associates with Galectin 3 and MHC class II molecules, TIM-3 pairs with Galectin-9 and CEACAM1, and BTLA engages HVEM; VISTA interacts with PSGL-1; B7-H3 (CD276) interacts with TREM-LT; and TIGIT binds to CD155 (PVR) and CD112 (PVRL2). Abbreviations: PD-1, Programmed Death-1; PD-L1, Programmed Death-Ligand 1; PD-L2, Programmed Death-Ligand 2; CTLA-4, Cytotoxic T-Lymphocyte-Associated Protein 4; LAG-3, Lymphocyte-Activation Gene-3; TIM-3, T-cell Immunoglobulin and Mucin-domain containing-3; BTLA, B and T Lymphocyte Attenuator; HVEM, Herpesvirus Entry Mediator; VISTA, V-domain Ig Suppressor of T-cell Activation; PSGL-1, P-selectin glycoprotein ligand-1; TREM-LT, Triggering receptor expressed on myeloid cells (TREM)-like transcript (LT); TIGIT, T-cell Immunoreceptor with Ig and ITIM domains; PVR, Poliovirus Receptor; PVRL2, Poliovirus Receptor-like 2.

**Figure 2 cancers-17-00880-f002:**
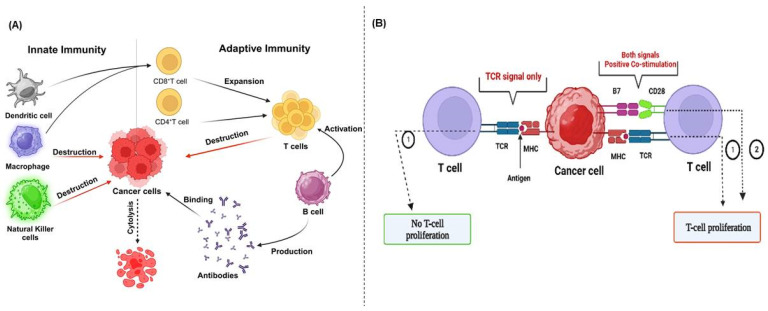
Interaction between innate and adaptive immunity in response to tumor cells. (**A**) Once tumor cells are identified, DCs and macrophages conduct phagocytosis of tumor cells. They also serve as APCs, presenting tumor antigens as a component of the MHC complex on their membranes to activate T cells. T cells eradicate tumor cells. NK cells initiate the process of destroying tumor cells through direct interactions. B cells can trigger T-cell activation and perform APC functions. B cells secrete antibodies that mediate ADCC and ADCP. (**B**) Signaling cascade from the interactions of tumor cells with naïve T cells. T-cell activation and proliferation necessitate the presence of two essential signals. The initial signal is initiated when a TCR engages with an antigen displayed on the surface of a tumor cell via MHC. Without a co-stimulatory receptor, T cells either undergo deletions or become anergic (nonfunctional). The second signal occurs when CD28 receptors on T cells interact with B7 proteins found on tumor cells. These combined signals are pivotal for initiating T-cell activation and subsequent proliferation. Abbreviations: DCs, dendritic cells; APCs, antigen-presenting cells; MHC, major histocompatibility complex; ADCC, antibody-dependent cellular cytotoxicity; ADCP, antibody-dependent cellular phagocytosis; TCR, T-cell receptor.

**Figure 3 cancers-17-00880-f003:**
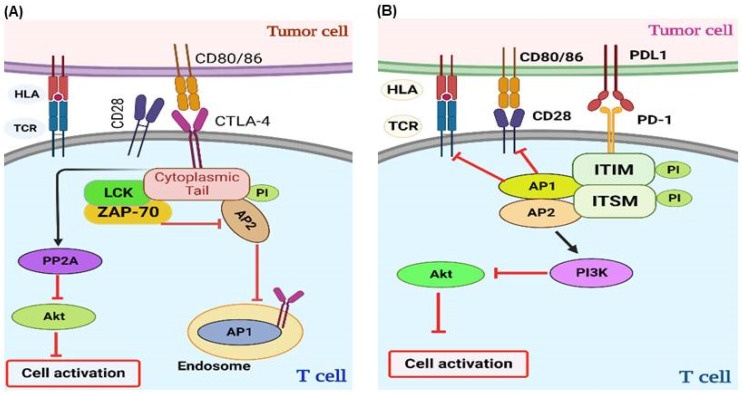
Mechanisms of immune-checkpoint signaling. (**A**) Mechanism of the CTLA-4 signaling pathway. Upon TCR engagement, intracellular vesicles containing CTLA-4 relocate to the immune synapse. Lck and ZAP-70 phosphorylate the cytoplasmic tail of CTLA-4, disrupting its intracellular transport by interfering with the interaction of AP-2. CTLA-4 inhibits T-cell activation by activating PP2A, which inhibits Akt signaling. (**B**) Mechanism of the PD-1 signaling pathway. PD-1 is phosphorylated at tyrosine residues within ITIM and ITSM on its cytoplasmic tail following TCR stimulation. Subsequently, it recruits phosphatases SHP-1 and SHP-2, which further dephosphorylate proximal signaling molecules downstream of TCR and CD28. PD-1 exerts its inhibitory effect on T-cell activation by activating PI3K via SHP-2, which inhibits Akt signaling. Abbreviations: TCR, T-cell receptor; HLA, human leukocyte antigen; mAb, monoclonal antibody; Lck, lymphocyte-specific protein tyrosine kinase; ZAP-70, ζ-chain-associated protein kinase 70; PP2A, protein phosphatase 2A; Pi, phosphorylation; AP2, activator protein 2; ITIM, immunoreceptor tyrosine-based inhibition motif; ITSM, immunoreceptor tyrosine-based switch motif; PI3K, phosphoinositide 3-kinase.

**Figure 4 cancers-17-00880-f004:**
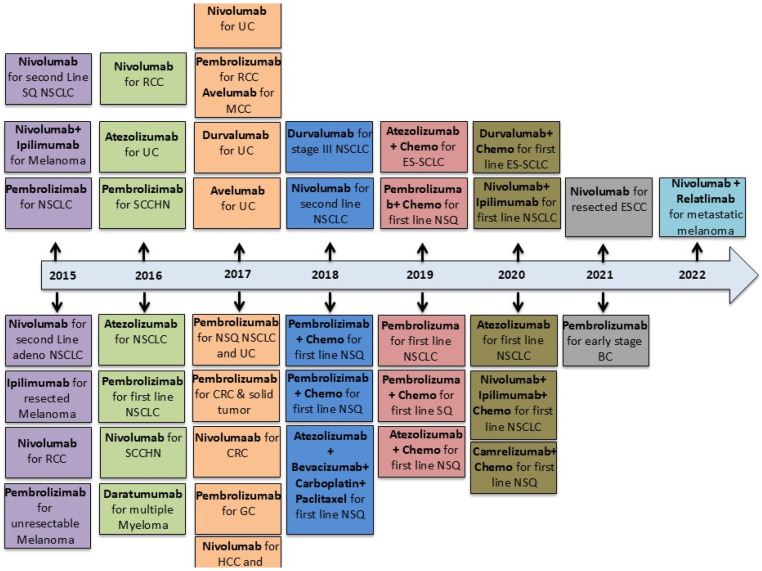
Showing the development of various immune checkpoint inhibitors for various cancer treatments over time. Abbreviations: RCC, Renal cell carcinoma; HNSCC, Head and Neck Squamous Cell Carcinoma; HCC, Hepatocellular carcinoma; ESCC, Esophageal squamous cell carcinoma; NSCLC, non-small cell lung carcinoma; GC, Gastric carcinoma; CC, cervical cancer; UC, Urothelial carcinoma; TNBC, Triple-negative breast cancer; SCLC, Small Cell Lung Cancer; MCC, Merkel cell carcinoma; CRC, Colorectal carcinoma.

**Figure 5 cancers-17-00880-f005:**
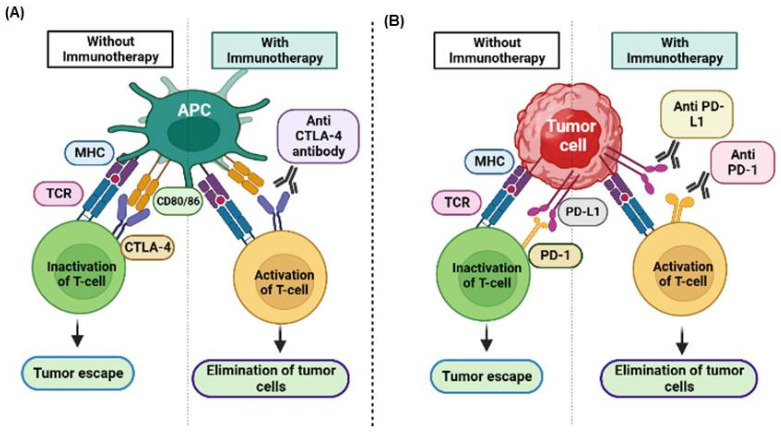
Ligand–receptor interactions between tumor cells and activated T cells and targets for anti-PD-1 and anti-CTLA-4 therapy. T-cell activation follows sequential progression, which is typically regulated by normal immune control mechanisms. Therapeutic interventions using anti-CTLA-4, anti-PD-1, and anti-PD-L1 antibodies have been designed to disrupt this regulation, leading to beneficial outcomes. (**A**) The interaction between the CTLA-4 receptor on T cells and the CD-80 ligand (B-7 homolog) on antigen-presenting cells promotes tumor immune evasion. When an anti-CTLA-4 antibody binds to CTLA-4, it enhances T-cell activation and enables the elimination of tumor cells. (**B**) The interaction between the PD-1 receptor on T cells and the PD-L1 ligand on tumor cells results in T-cell dysfunction and tumor immune evasion. In the presence of an anti-PD-1 or anti-PD-L1 antibody, T cells are reactivated, initiating the death of tumor cells. Abbreviations: CTL-4, T-lymphocyte-associated antigen 4; PD-1, programmed cell death 1; MHC, major histocompatibility complex; TCR, T-cell receptor.

**Figure 6 cancers-17-00880-f006:**
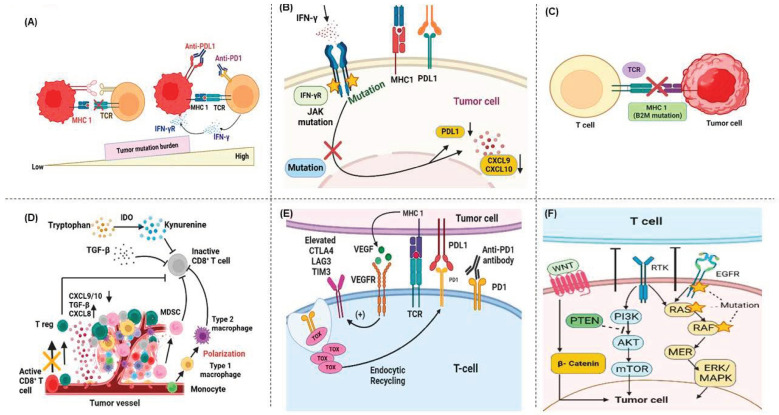
Mechanisms of primary resistance. (**A**) Tumors characterized by a high mutation burden usually exhibit a more favorable response to anti-PD-1/PD-L1 therapy because they are more likely to generate immunogenic neoantigens. These neoantigens activate CD8^+^ T cells and stimulate a robust antitumor immune response. (**B**) Tumor cells that have developed resistance to IFN-γ signaling due to primary JAK1/2 mutations may not induce PD-L1 upregulation but can still inhibit T-cell reactivity through PD-1/PD-L1-independent pathways. In addition, inactivation of IFN-γ signaling leads to reduced expression of CXCL9 and CXCL10, which are critical for T-cell recruitment. (**C**) Tumor cells with abnormal expression of antigen presentation pathway components fail to effectively present tumor antigens, thus hindering the elicitation of antitumor immunity required to eliminate cancer cells. (**D**) Within the TME, a diverse array of immunosuppressive cells can affect the efficacy of anti-PD-1/PD-L1 therapy by suppressing T-cell reactivity. Cytokines produced by tumors attract more immunosuppressive cells into the TME and promote their polarization toward a pro-tumor phenotype. (**E**) Alternative immune-checkpoint molecules are upregulated in T cells infiltrating the tumor. This upregulation, coupled with increased VEGFR signaling and TOX expression, exacerbates the activation of inhibitory signaling pathways. (**F**) Mutations in oncogenes and aberrant activation can thwart the development of an effective antitumor immune response, leading to primary resistance to immunotherapy. Abbreviations: CTL-4, T-lymphocyte-associated antigen 4; CXCL, chemokine motif (C-X-C) L ligand; IFN-γ, interferon-gamma; IFN-γ R, interferon-gamma receptor; IDO, indoleamine 2,3-dioxygenase; JAK, Janus kinase; LAG-3, lymphocyte-activation gene 3; MHC, major histocompatibility complex; MDSC, myeloid-derived suppressive cells; MAPK, mitogen-activated protein kinase; PD-1, programmed cell death 1; PD-L1, programmed death-ligand 1; PTEN, phosphatase and tensin homolog; PI3K, phosphatidylinositol 3-kinase; TOX, thymocyte selection-associated high-mobility group bOX; TCR, T-cell receptor; TGF, transforming growth factor; TIM-3, T-cell immunoglobulin and mucin-domain 3; TME, tumor microenvironment; VEGF, vascular endothelial growth factor; β2M, beta-2 microglobulin; APP, antigen processing and presentation; TAPs, transporters associated with neoantigen presentation.

**Figure 7 cancers-17-00880-f007:**
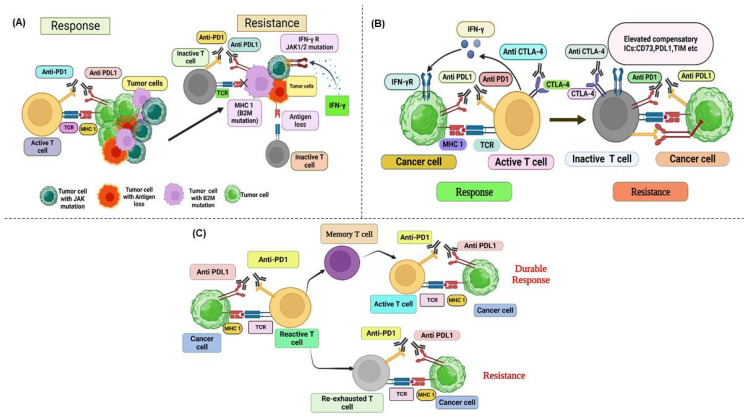
(**A**,**B**). Monoclonal antibodies (mAbs) have become powerful tools in cancer treatment. Notably, immune checkpoint inhibitors (ICIs) like anti-PD-L1 and anti-PD-1 mAbs have shown significant effectiveness against multiple cancers through TCRs and MHC class I. Tumor cells often resist immune checkpoint inhibitors (ICIs) due to high mutation rates in MHC class I and JAK1/2, which impair immune recognition. (**C**). Positive responses to immune checkpoint inhibitors (ICIs) are linked to increased levels of specific T lymphocyte subsets, like memory T cells. However, prolonged exposure to tumor antigens and an immunosuppressive tumor microenvironment (TME) can lead to T-cell exhaustion. Despite this, immunotherapy has been shown to trigger lasting immune responses, which can continue even after treatment ends, leading to extended antitumor effects and improved overall survival. The process of immunoediting, driven by the pressure exerted through PD-1/PD-L1 blockade, usually favors the survival of tumor cells with a heightened capacity to evade the antitumor immune response. As therapy progresses, compensatory inhibitory signaling pathways are activated, making it challenging for the PD-1/PD-L1 and CTLA-4 blockade to effectively re-energize CD8^+^ T cells. If tumor-specific T cells fail to transition into memory T cells, the treatment response is sustained, potentially leading to disease recurrence or acquired resistance following discontinuation of therapy. Abbreviations: CTL-4, T-lymphocyte-associated antigen 4; IFN-γ, interferon-gamma; JAK, Janus kinase; MHC, major histocompatibility complex; TCR, T-cell receptor; TIM-3, T-cell immunoglobulin and mucin-domain 3; β2M, beta-2 microglobulin; ICs, immune checkpoints.

**Table 2 cancers-17-00880-t002:** Strategies used to overcome Resistance to ICIs.

Mechanism	Type of Strategy	Example	Ref.
Enhancing T-Cell Infiltration	Co-stimulatory agonist	Ab-guided LIGHT fusion protein + anti-PD-L1 Ab	[352]
Overcoming T-cell exhaustion	Blockade of alternate co-inhibitory ICIs	Nivolumab + Anti-TIM3 blocking Ab	[183,184]
		Anti-TIGIT + Anti-PD-1 Ab	[184]
	Co-stimulatory agonists including 4-1BB, OX40, CD40, GITR, and ICOS	Anti-PD-1 Ab + Agonistic anti-CD40 Ab	[353]
Inhibition of epigenetic modifiers	Epigenetic regulation	DNA methyltransferase inhibitor and histone deacetylase inhibitors + anti-PD-1 and anti-CTLA-4 Ab	[354]
Improving the Tumor Immune Microenvironment	Cytokine or chemokine receptor blockade	Anti-PD-1 Ab + CSF1R blockade	[273]
		Anti-PD-1 Ab + CXCR2 mAb blockade	[282]
		CXCR4 mAb blockade	[292]
	IDO inhibitor	Anti-PD-L1 & anti- CTLA-4 Ab + IDO inhibitor	[355]
	Inhibition of adenosinergic pathway	Anti-PD-1 Ab + CD73 inhibitor	
		Anti-PD-1 Ab + A2AR antagonist	[356]
	Inhibition of TGF-β	Anti-PD-1 Ab + TGF-β blockade	[357]
	VEGF inhibitors	Bevacizumab + atezolizumab + chemotherapy	[115]
	PI3K inhibitor	Anti-CTLA-4 Ab + Anti-PD-1 Ab + Selective PI3K blockade	[358]
T-Cell Priming Enhancement	Cancer vaccines	Nivolumab + Multipeptide vaccine	[359]
		Ipilimumab + T-VEC	[146]
		Pembrolizumab + T-VEC	[147]
	Chemotherapy	Carboplatin & paclitaxel + Pembrolizumab	[360]
	Radiotherapy	Pembrolizumab + radiotherapy	[361]
	TLR agonist	Anti-PD-L1 Ab + TLR3-RNA agonist	
		Anti-PD-L1/PD-1 Ab + TLR9 agonist	[362]
	IFN-α	anti-PD-L1-IFN-α fusion protein	[132]
Combination with other therapies	Oncogenic pathway inhibitor	Targeting MAPK pathway with BRAF inhibitor (vemurafenib) + Anti-PD-1/PD-L1 Ab	[363]
		BRAF, ERK inhibitors + Anti-PD-1 Ab	[364]

Abbreviations: LIGHT, which stands for homologous to Lymphotoxin, exhibits inducible expression and competes with HSV glycoprotein D for binding to herpesvirus entry mediator, a receptor expressed on T lymphocytes; PD-L1, Programmed death-ligand 1; PD-1, Programmed cell death protein 1; TIM3, T-cell immunoglobulin and mucin-domain-containing protein 3; TIGIT, T-cell immunoglobulin and ITIM domain; Ab, antibody; ICOS, Inducible Co-Stimulator; GITR, Glucocorticoid-Induced TNFR-Related protein; CTLA-4, Cytotoxic T Lymphocyte-Associated molecule-4; (CSF1R), Colony-stimulating factor 1 receptor; (CXCR2), C-X-C chemokine receptor 2; (IDO), Indoleamine 2,3-dioxygenase; A2A, Adenosine Receptor 2A; (TGF-β), Transforming growth factor-β; VEGF, Vascular endothelial growth factor; PI3K, phosphatidylinositol-3 kinase; T-VEC, Talimogene laherparepvec; TLR, Toll-like receptor; IFN-α, Interferon alpha; MAPK, Mitogen-activated protein kinase; mAbs, monoclonal antibodies.

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
