# Peer review of "Revolutionary Cancer Therapy for Personalization and Improved Efficacy: Strategies to Overcome Resistance to Immune Checkpoint Inhibitor Therapy"

_cancers, 2025, doi:10.3390/cancers17050880_

Round 1
Reviewer 1 Report
Comments and Suggestions for Authors
This is a very well compiled and detail review that discuss some very apt issues. This reviewer would only recommend some minor editing.
1. Figure 4, resolution is low.
2. Figure 4 caption seems incomplete.
3. Page 16-17 line no 673-689, bold fonts used.
4. rechecking format of the manuscript advised
Author Response
This is a very well compiled and detail review that discuss some very apt issues. This reviewer would only recommend some minor editing.
Comment 1: Figure 4, resolution is low.
Response: Thank you for your comment. We have included the figure with high resolution.
Comment 2: Figure 4 caption seems incomplete.
Response: Thank you for your comment. We have dove necessary corrections based on your feedback.
Comment 3: Page 16-17 line no 673-689, bold fonts used.
Response: Thank you for your comment. In the revised manuscript, we have changed the bold fonts to normal style to ensure consistency and improve the overall readability of the text.
Comment 4: rechecking the format of the manuscript advised
Response: Thank you for your suggestion. We have thoroughly reviewed the manuscript and made the necessary changes to improve clarity, consistency, and overall quality. We appreciate your valuable feedback and believe the revisions enhance the manuscript.
Reviewer 2 Report
Comments and Suggestions for Authors
This review comprehensively explores the topic of immune checkpoint inhibitors and examines all aspects related to their clinical application, limitations, and future therapeutic developments to improve the efficacy of such therapies and circumvent primary and secondary resistance of tumors, while considering strategies to reduce adverse effects that often lead to discontinuation of therapy. This review serves as a comprehensive resource on immunocheckpoints, compiling information on the latest developments and challenges for such therapy. It is undoubtedly informative and valuable for educational purposes. However, from this reviewer's perspective, it would benefit from streamlining in certain areas to enhance readability. Therefore, the following recommendations are proposed:
1) The introduction is excessively lengthy and would benefit from concision by eliminating information that is described and expanded upon in subsequent paragraphs. This would prevent the repetition of certain concepts.
2) It is advisable to divide paragraph 2.1 into subsections to facilitate comprehension. For example: a) function of immunocheckpoints; b) mechanism of action of ICIs (other targets such as TIGIT, VISTA, etc. could be reported here); c) preclinical and clinical studies; d) strategies to improve efficacy of ICIs (steric hindrance; endosomal sorting complex, etc.)
3) The chapter on artificial intelligence is somewhat general in nature, being highly speculative, and its relevance to the purpose of the review is questionable.
4) It is recommended to review the article again to correct typographical errors, such as at line 670 (delete: These receptors). Additionally, it would be beneficial to better summarize certain concepts, such as that of trogocytosis, which is repeated unnecessarily between lines 252-256.
Author Response
This review comprehensively explores the topic of immune checkpoint inhibitors and examines all aspects related to their clinical application, limitations, and future therapeutic developments to improve the efficacy of such therapies and circumvent primary and secondary resistance of tumors, while considering strategies to reduce adverse effects that often lead to discontinuation of therapy. This review serves as a comprehensive resource on immunocheckpoints, compiling information on the latest developments and challenges for such therapy. It is undoubtedly informative and valuable for educational purposes. However, from this reviewer's perspective, it would benefit from streamlining in certain areas to enhance readability. Therefore, the following recommendations are proposed:
Comment 1: The introduction is excessively lengthy and would benefit from concision by eliminating information that is described and expanded upon in subsequent paragraphs. This would prevent the repetition of certain concepts.
Response: Thank you for your suggestion. We reviewed the introduction section, which was excessively lengthy, and removed unnecessary content. We eliminated information that is already described and expanded upon in subsequent paragraphs, improving the focus and clarity of the manuscript.
Comment 2: It is advisable to divide paragraph 2.1 into subsections to facilitate comprehension. For example: a) function of immunocheckpoints; b) mechanism of action of ICIs (other targets such as TIGIT, VISTA, etc. could be reported here); c) preclinical and clinical studies; d) strategies to improve efficacy of ICIs (steric hindrance; endosomal sorting complex, etc.)
Response: Thank you for your valuable advice to divide Section 2.1 into subsections for better comprehension. We have revised this section into two subsections: 2.1.1 " Immune Checkpoint Inhibitors (ICIs) and Mechanism of action " and 2.1.2 "Preclinical and Clinical Studies." Additionally, we deleted the brief portion on "Strategies to Improve Efficacy of ICIs," as expanding on this topic would require additional content that was not feasible to include in the current manuscript.
Comment 3: The chapter on artificial intelligence is somewhat general in nature, being highly speculative, and its relevance to the purpose of the review is questionable.
Response: Thank you for your advice. We agree that the chapter on artificial intelligence was somewhat general, and its relevance to the review’s purpose was limited. Based on your suggestion and input from other reviewers, we have removed that section in the revised manuscript.
Comment 4: It is recommended to review the article again to correct typographical errors, such as at line 670 (delete: These receptors). Additionally, it would be beneficial to better summarize certain concepts, such as that of trogocytosis, which is repeated unnecessarily between lines 252-256.
Response: We reviewed the article to correct typographical errors and streamlined the content by summarizing trogocytosis only once, eliminating the unnecessary repetition.
Reviewer 3 Report
Comments and Suggestions for Authors
The author provided a comprehensive review of immune system and immunotherapy, combination of immunotherapies, immune-related adverse events, resistance to immunotherapy, therapeutic strategies for overcoming drug resistance with ICIs, comprehensive biomarkers in ICI therapy, microbiome and immunotherapy, and artificial intelligence and immunotherapy.
In the introduction, this author states that “the review will focus on the most widely recognized ICs and the effect of their inhibition on the immune response. Furthermore, it addresses the clinical challenges faced in the clinic, including treatment resistance and the occurrence of irAEs”. The journal (Cancers) 's instructions to authors of review articles state, “These should provide a concise and accurate update on the latest advances made in a particular area of research”. The article covered various aspects of immunotherapy in a similar tone, resulting in an overly long paper, but it did not provide a concise update. The review needs to clearly state what the author's main message to the reader is as the latest advances. The author appears to focus on resistance to ICI and strategies to overcome it. This point should be expressed in the title and emphasized at the end of the introduction. In addition, repeated explanations and less relevant items need to be removed. The overall volume of the paper needs to be reduced. This paper is not acceptable in its current form.
The storyline of this paper proceeds in the following order: 2. Immune system, 3. Combination of immunotherapies, 4. IrAEs, 5. Resistance to immunotherapy, 6. Therapeutic strategies, 7. Comprehensive biomarkers in ICI therapy 8. Microbiome, 9. AI. However, for example, the order of sections could be changed to 2, 7, 8, 4, 5, 6. Section 3 could be deleted. Also, much of the content of subsections 2.2 Cytokines, 2.3 Cancer Viruses, 2.4 ACT, and 2.5 Bispecific ICIs is repeated in Section 6 and could be deleted. Section 9 can be deleted; AI is a promising research area in cancer immunotherapy, but the number of published ICI-related papers is still too limited to review. Instead, AI can be briefly described in the Future Perspectives section.
The title contains somewhat figurative language that needs to be changed to better describe the content of the paper. For example, “Revolutionary Cancer Therapy for Personalization and Improved Efficacy: Strategies to Overcome Resistance to Immune Checkpoint Inhibitor Therapy”.
Abstract: The journal requires a simple summary (no more than 150 words) and an abstract (250 words) following the instructions of the authors of the article or systematic review. Therefore, for the current abstract, it should be short.
Figure 4: There is no explanation of Figure 4 in the text. This figure needs to be deleted.
Figure 7 legend: explanatory text is needed for each of (A) through (C).
The authors repeat abbreviations in many places. For example, programmed death-ligand 1 (PD-1) on page 2, line 68 and page 3, line 101; cytotoxic T-lymphocyte-associated antigen-4 (CTLA-4) on page 2, line 72, and page 3, line 101. This repetition of words and abbreviations is common throughout the paper and should be corrected.
Page 33, line 1387: “Cancer Vaccines” would be “Oncolytic Virus Therapies and Vaccines Against Cancers”.
The introduction and conclusion are also long. They need to be reduced to less than half.
References 205 and 210 are the same. Need to check for other papers in the references.
Author Response
The author provided a comprehensive review of immune system and immunotherapy, combination of immunotherapies, immune-related adverse events, resistance to immunotherapy, therapeutic strategies for overcoming drug resistance with ICIs, comprehensive biomarkers in ICI therapy, microbiome and immunotherapy, and artificial intelligence and immunotherapy.
Comment 1: In the introduction, this author states that “the review will focus on the most widely recognized ICs and the effect of their inhibition on the immune response. Furthermore, it addresses the clinical challenges faced in the clinic, including treatment resistance and the occurrence of irAEs”. The journal (Cancers) 's instructions to authors of review articles state, “These should provide a concise and accurate update on the latest advances made in a particular area of research”. The article covered various aspects of immunotherapy in a similar tone, resulting in an overly long paper, but it did not provide a concise update. The review needs to clearly state what the author's main message to the reader is as the latest advances. The author appears to focus on resistance to ICI and strategies to overcome it. This point should be expressed in the title and emphasized at the end of the introduction. In addition, repeated explanations and less relevant items need to be removed. The overall volume of the paper needs to be reduced. This paper is not acceptable in its current form.
The storyline of this paper proceeds in the following order: 2. Immune system, 3. Combination of immunotherapies, 4. IrAEs, 5. Resistance to immunotherapy, 6. Therapeutic strategies, 7. Comprehensive biomarkers in ICI therapy 8. Microbiome, 9. AI. However, for example, the order of sections could be changed to 2, 7, 8, 4, 5, 6. Section 3 could be deleted. Also, much of the content of subsections 2.2 Cytokines, 2.3 Cancer Viruses, 2.4 ACT, and 2.5 Bispecific ICIs is repeated in Section 6 and could be deleted. Section 9 can be deleted; AI is a promising research area in cancer immunotherapy, but the number of published ICI-related papers is still too limited to review. Instead, AI can be briefly described in the Future Perspectives section.
Response: Thank you for taking the time to thoroughly review our manuscript and providing valuable suggestions to improve it. Based on your feedback, we have removed the AI section and reflected "Resistance to ICI and Strategies to Overcome It" in the title. We have also emphasized this section at the end of the introduction. Additionally, we made the necessary changes to correct the mistakes you identified. We appreciate your insightful contributions.
Comment 2: The title contains somewhat figurative language that needs to be changed to better describe the content of the paper. For example, “Revolutionary Cancer Therapy for Personalization and Improved Efficacy: Strategies to Overcome Resistance to Immune Checkpoint Inhibitor Therapy”.
Response: Thank you for your valuable suggestion. We appreciate your input and have revised the title to: “Revolutionary Cancer Therapy for Personalization and Improved Efficacy: Strategies to Overcome Resistance to Immune Checkpoint Inhibitor Therapy” as recommended.
Comment 3: Abstract: The journal requires a simple summary (no more than 150 words) and an abstract (250 words) following the instructions of the authors of the article or systematic review. Therefore, for the current abstract, it should be short.
Response: Thank you for your suggestion. We reviewed the abstract of our manuscript and removed some content. We adjusted the word count as per your recommendation, improving the focus and clarity of the abstract for a more concise presentation.
Comment 4: Figure 4: There is no explanation of Figure 4 in the text. This figure needs to be deleted.
Response: Thank you for your suggestion. However, we feel the figure 4 is important and we have included it in the text and included details as well.
Comment 5: Figure 7 legend: explanatory text is needed for each of (A) through (C).
Response: Thank you for your comment. We have added an explanatory figure legend to Figure 7 in the revised manuscript as A,B. Monoclonal antibodies (mAbs) have become powerful tools in cancer treatment. Notably, immune checkpoint inhibitors (ICIs) like anti-PD-L1 and anti-PD-1 mAbs have shown significant effectiveness against multiple cancers through TCR and MHC class I. Tumor cells often resist immune checkpoint inhibitors (ICIs) due to high mutation rates in MHC class I and JAK1/2, which impair immune recognition. C. Positive responses to immune checkpoint inhibitors (ICIs) are linked to increased levels of specific T lymphocyte subsets, like memory T cells. However, prolonged exposure to tumor antigens and an immunosuppressive tumor microenvironment (TME) can lead to T-cell exhaustion. Despite this, immunotherapy has been shown to trigger lasting immune responses, which can continue even after treatment ends, leading to extended antitumor effects and improved overall survival
Comment 6: The authors repeat abbreviations in many places. For example, programmed death-ligand 1 (PD-1) on page 2, line 68 and page 3, line 101; cytotoxic T-lymphocyte-associated antigen-4 (CTLA-4) on page 2, line 72, and page 3, line 101. This repetition of words and abbreviations is common throughout the paper and should be corrected.
Response: Thank you for pointing out these mistakes. We have corrected the repeated abbreviations in the revised manuscript to ensure consistency and clarity throughout the text.
Comment 7: Page 33, line 1387: “Cancer Vaccines” would be “Oncolytic Virus Therapies and Vaccines against Cancers”.
Response: Thank you for your suggestion. We have accepted your recommendation and replaced "Cancer Vaccines" with "Oncolytic Virus Therapies and Vaccines Against Cancers" in the revised manuscript
Comment 8: The introduction and conclusion are also long. They need to be reduced to less than half.
Response: Thank you for your suggestion. We have removed some content from both the introduction and conclusion sections and have restructured them to improve clarity and ensure a more concise, well-organized presentation
Comment 9: References 205 and 210 are the same. Need to check for other papers in the references.
Response: Thank you for your comment. We have corrected References 205 and 210 in the revised manuscript.
Round 2
Reviewer 3 Report
Comments and Suggestions for Authors
The author respond appropriately to the points raised by the reviewer.